# C1-linker region of PARG1 RhoGAP promotes the catalytic recognition fold of RhoA substrate

Zen Kouchi[1]*, Masaki Kojima[2]

**1** Department of Neuronal Information, Institute for Developmental Research, Aichi Developmental Disability Center, Kasugai Aichi, Japan, **2** School of Life Sciences, Tokyo University of Pharmacy and Life Sciences, Hachioji Tokyo, Japan

\* zkouchi@inst-hsc.jp

## Abstract

PARG1 (ArhGAP29) belongs to a class of F-BAR proteins that contain a GTPase activating (GAP) domain that stimulates the GTP-to-GDP conversion of RhoGTPases. In this study, the substrate-recognition mechanism of human PARG1 was structurally modeled in computational approaches. Docking analysis using HDOCK showed that the predicted RhoGAP domain containing the N-terminal C1 region harbored structural determinants only for RhoA recognition with its catalytic loop and the $\alpha_4$- and α9–10 helices of the GAP domain. Molecular dynamics of wild-type PARG1-RhoA complex revealed that the predicted C1 structure depicted unique interface for the $\alpha_3$ helix of RhoA, leading to stable interaction with the RhoA substrate. Interestingly, RhoA interacted with the C1-GAP domains with missense mutations such as p.Thr622Met (T622M) and p.Ile845Val (I845V) differently, but the several interface residues in the catalytic loop and C-terminal $\alpha_9$-$\alpha_{10}$ helices were not matched to the known crystallized complexes in molecular dynamics simulation. PARG1 I845V mutant complex was theoretically deduced to disorganize RhoA interfaces and T622M mutation decreased the substrate affinity to 80% of that of WT PARG1 complex. The C-terminal C1 domain that formed a coiled-coil structure in a wild-type specific manner and the loop regions adjacent to the GAP region modulated the corresponding C1 interaction interfaces in RhoA. There were differences in motions of the conserved and variable interface residues among RhoGAP domains that locate in the $\alpha_{9–10}$ loop and C-terminal $\alpha_4$ and N-terminal $\alpha_{9–10}$ helices of the GAP domain between WT and mutant RhoGAP-RhoA complexes. The stable RhoA interaction specific to wild-type PARG1 is attributed to the motions of the GAP region including the C1 domain, in contrast to mutant PARG1 GAP domains that tended to disorganize the catalytic complex.

**Data availability statement:** All relevant data are within the manuscript and its Supporting Information files.

**Funding:** The author(s) received no specific funding for this work.

**Competing interests:** The author has declared that no competing interests exist.

## Introduction

RhoGTPases are regulated by RhoGAP and Rho guanine nucleotide exchange factors (RhoGEFs), which spatiotemporally regulate GTP-hydrolyzing activities and catalysis of the exchange of GDP with GTP, respectively, and Rho guanine dissociation inhibitors (RhoGDI), which inhibits exchange of bound GDP for GTP [1]. Local RhoGTPase activation is controlled by large molecular complexes or RhoGAPs, which generally lack substrate selectivity and are regulated by specific domain modules, including the C1 and C2 domains [2].

PARG1 (also known as ArhGAP29) possess an N-terminal F-BAR/BAR domain and belongs to the C1 domain-containing RhoGAP family [2,3]. Mutations in PARG1 increase the risk of non-syndromic cleft lip and/or palate (NSCL/P), a common birth defect with a complex etiology [4–7]. Several missense, nonsense, and frameshift mutations that cause NSCL/P have been identified in the entire coding region of PARG1 [4,7]. However, the effects of mutations on the catalytic domain and enzymatic regulation remain elusive, although a loss-of-function paradigm has been suggested [4]. p.Thr622Met and p.Arg616His variations reside within the C1 domain, and the former and a missense variant p.Ile845Val in the GAP domain were identified in individual with NSCLP [4]. pArg616His was found in both Filipino cases and controls. It was reported that phenotypes of *Arhgap29* null mice and knock-in mice coding a rare variant, p.Lys326X which generates truncated transcripts, are embryonic lethal, while functional effects of other variants on embryonic development remains unknown in model organisms [6]. Recent genetic studies have shown that IRF6-mediated signaling is important for craniofacial development, and studies on downstream factors regulating developmental cell adhesion and cytoskeletal organization have focused on elucidating the etiology of NSCL/P [4,8,9]. Although PARG1 is specific to RhoA [3,10], the structure of RhoGAP and its RhoA complex have not yet to be elucidated.

The preference of PARG1 GAP activity towards RhoA was reported using the recombinant GAP domain protein with the N- and C-terminal regions defined by InterPro [3]. FRET-based screening of substrates and proximity interactome analysis using constitutively active RhoGTPases revealed GAP activity of PARG1 towards RhoA [10,11]. On the other hand, in the case of ArhGAP35, RhoA family proteins (RhoA, B, and C) were not detected as its substrates in proximity labeling method [4], but they are selected as the GAP substrates using cell-based or enzymatic assays [2,10]. Interestingly, the catalytic GAP domain of GMIP containing the C1 domain was highly active on RhoA [10,12]. As for the other C1 domain-containing RhoGAPs, the GAP activity of the C1-RhoGAP region of N-chimerin is negatively regulated by the N-terminal SH2 domain, and it translocates to the plasma membrane through a C1 region-mediated interaction with diacylglycerol upon activation [10,13,14]. The N-terminal BAR domain of HMHA1 also plays a negative regulatory role in RhoGAP activity, and ectopic expression of N-terminal deletion mutants that exhibit GAP activity induces significant focal adhesion loss in HeLa cells [15]. The RhoGAP C1 domain has low sequence homology with the corresponding regions of chimerin and HMHA1. These results suggest that several domains adjacent to GAP region intrinsically

modulate the specificity or binding affinity to GTPases. Especially, F-BAR RhoGAPs containing the C1 domain may catalyze their substrates *via* a catalytic fold composed of the C1 and RhoGAP domains in diverse signaling pathways.

In this study, the RhoA recognition mechanism of human PARG1 was structurally deduced, and the GAP domain and C1-loop region were selected as the catalytic unit for RhoA. Interestingly, molecular dynamics (MD) simulation studies showed that the flexible C1-linker domain of wild-type (WT) PARG1 functioned as a RhoA interface, leading to a stable interaction with the inactive form of RhoA possessing flexible switch I region, but the RhoA-bound mutant p.Thr622Met (T622M) and p.Ile845Val (I845V) PARG1 complexes showed distinct motions of the substrates with disorganized interface residues that were not matched to the residue-residue contacts of the crystallized complex structures. The deduced catalytic models revealed strict regulation of the substrate specificity and binding properties of PARG1, and how missense mutations affect its catalytic fold revealed the putative critical residues for structural recognition of the RhoA substrate by the C1-GAP domain.

## Materials and methods

### Structural modeling

Sequence of human PARG1 (UniProt ID: Q52LW3) retrieved from UniProt was submitted to HHPred for analyzing the secondary structures, and multiple alignment-based structural modeling was performed using a SWISS-MODEL search [16–18]. A suitable model was evaluated using trRosetta [19] and I-Tasser [20], which ranked at the top in the CASP, to check the potential templates [21–23]. The potential templates of the GAP domain retrieved from Protein Data Bank (PDB) [24] were N-chimerin (3cxl) and MgcRacGAP (5c2k) (S1 Table). Predicted models were evaluated using PROCHECK, ERRAT, and ProSA programs (S2 Table) [21,22,25] and the structural assessment identified N-chimerin as the best template (S2 Table). Structural model of RhoGAP of PARG1 containing loop region and the C1 domain were also predicted using HHPred and trRosetta [17,19]. A structural evaluation using these programs selected model 4 for the GAP with N- and C-terminal loop regions and model2 for the GAP domain containing the C1 domain as the best model, respectively (S2 Table, Fig 1c–1e). The RhoGAP domains of T622M and I845V missense mutants were also modeled using these programs. The predicted C1-RhoGAP models were also evaluated using PROCHECK, ERRAT, and ProSA programs, and the structural models were used for docking analysis and molecular dynamics [21,22,25]. The details in the missense mutations are described in HGMD database (https://www.hgmd.cf.ac.uk/ac/index.php) [4].

### Molecular docking analysis

Docking analysis of the modeled structure interacting with RhoGTPases was performed using HDOCK in the template-free docking mode [26]. In template-free mode, possible binding modes are sampled through fast-Fourier transform-based calculation with a shape complementarity scoring method. The fixed receptor and rotated ligand in rational 3D Euler space with an angle interval of 15º are mapped onto grids by considering the long-range interactions of atoms. In every rotation of the ligand, top orientations with the best shape complementarity scores are evaluated by a scoring function, ITScorePP, and the top-score is ranked based on the binding energy scores and clustering. The inter-atom energies and side chains of the modeled structures were optimized using EGAD program [27]. The ability of the program to reproduce the RhoA- or Cdc42-RhoGAP interactions was evaluated using either 5c2k or 5c2j, respectively. or via the modeled N-chimerin complex (PDB code: 3cxl) with Rac1, which showed similar tertiary structure toward each RhoGTPase using the HDOCK program [23]. The following structures of RhoGTPases were used for the docking analysis: RhoA (PDB code: 5c2k, 6r3v, 6v6m, 1a2b, 1ow3, 3msx, and 5irc), Cdc42 (PDB codes: 5c2j, 1grn, and 2ngr), Rac1 (PDB code: 3th5), RhoB (PDB code: 6sge), RhoC (PDB code: 1z2c), RhoD (PDB code: 7kdc), RhoG (PDB code: 6uka), and Tc10 (PDB code: 2atx). As for the modeled complex using AlphaFold software, the corresponding domain of PARG1 (http://alphafold.ebi.ac.uk) was used for HDOCK analysis or the selected complexes using AlphaFold-multimer were selected for deduced structure of RhoA-bound C1-GAP domain, based on RMSD values to the crystallized GAP complexes [28,29]. *in silico* evaluation of alanine

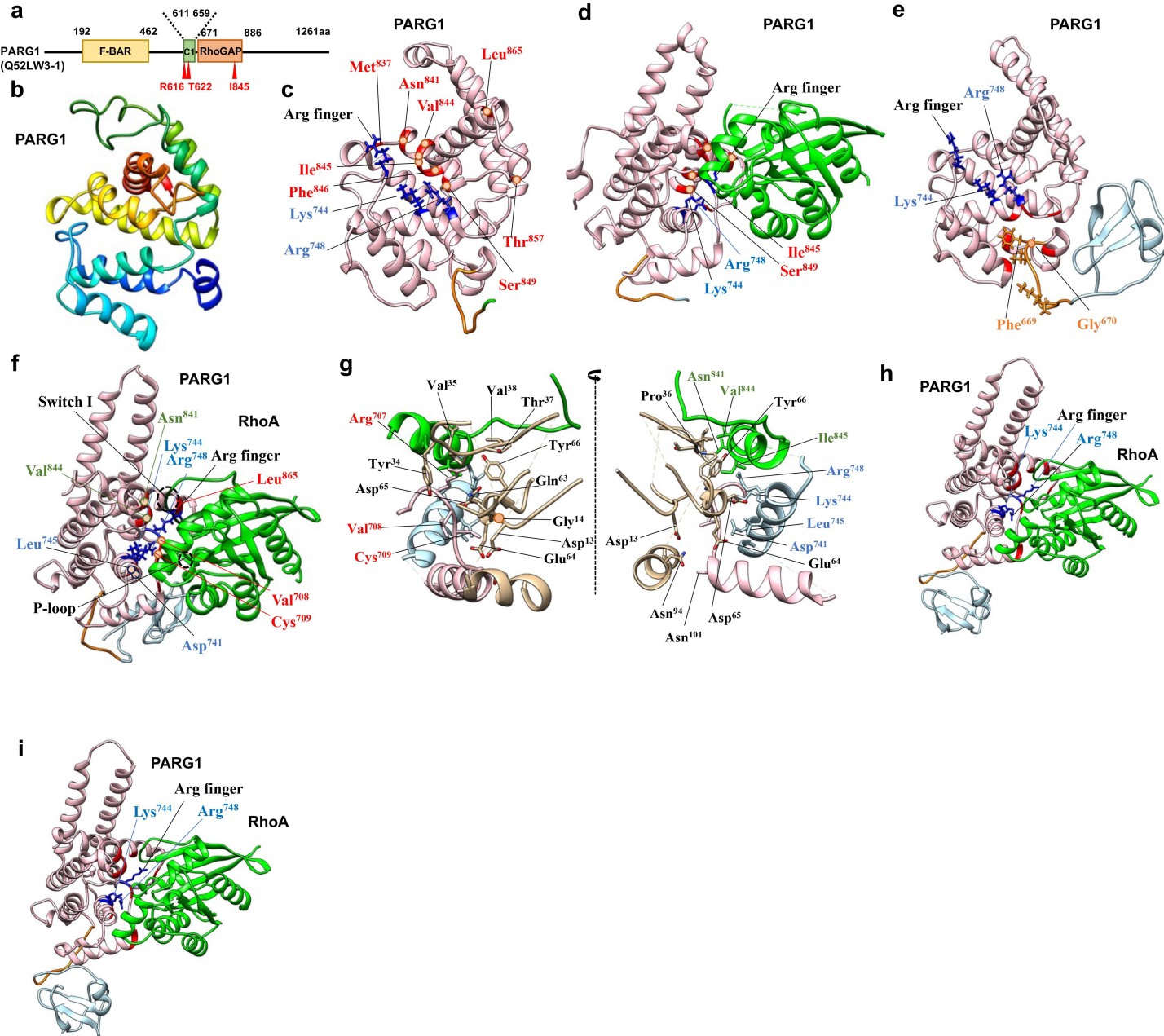

**Fig 1. Modeled structure of RhoGAP domain and C1-RhoGAP domains of PARG1.** (**a**) Schematic view of domain organization of *Homo sapiens* PARG1. (**b**) Modeled structure of PARG1 RhoGAP domain without the C1-linker domain was deduced based on SWISS-MODEL. (**c**) The PARG1 (residue number: 658-898) was structurally modeled using HHPred and trRosetta programs. The conserved interface residues with 5irc are indicated as blue for positively-charged interface and as red for C-terminal interface of RhoA interaction, respectively. (**d**) The modeled RhoA-bound PARG1 structure (residue number: 658-898) was selected using HDOCK with template-free mode and the structural evaluation of the complexes was performed using Drugscore^PPI. The side chains of positively charged interface residues and the arginine finger (Arg^707) are indicated on their location marked with blue. The positions of predicted C-terminal interface residues are indicated as red in the cartoon. (**e**) The PARG1 RhoGAP with the N-terminal C1 domain (residue number: 611-886) was structurally modeled using HHPred and trRosetta programs, and depicted as the N-terminal C1 domain (green) and RhoGAP domain (pink) with connecting linker region (orange). (**f**) The modeled RhoA-bound structure was selected using HDOCK with template-free mode. The structural evaluation of the complexes was performed using Drugscore^PPI. The side chains of positively charged interface residues and the arginine finger (Arg^707) are indicated on their location marked with blue. The positions of predicted C-terminal interface residues are indicated as red in the cartoon. (**g**) The side chains of PARG1 RhoGAP interface for RhoA (< 6Å in distance) were structurally modeled. The residues in the p-loop and switch I, II regions

of RhoA (black letters with gold ribbon) interacted with the catalytic interface: catalytic arginine loop, pink; the positively charged interface, light blue; C-terminal interface, green. (**h-i**) The modeled RhoA-bound structure using ColabFold based on AlphaFold prediction. The structural evaluation of the complex model 1 (**h**) and 3 (**i**) was performed using Drugscore[PPI]. The side chains of positively charged interface residues and the arginine finger (Arg[707]) are indicated on their location marked with blue. The positions of predicted interface residues (>0.5 in Drugscore[PPI] score) are indicated as red in the cartoon.

scanning mutagenesis using the DrugScore[PPI] simulation was performed for binding interface analysis and used for the selection of RhoA-PARG1 complexes [2,30] (S1 Fig). As for experimentally validated RhoGAP-RhoGTPase complex, docking of each p50GAP and Cdc42 structure using HDOCK with template-free mode selected the well-matched interface residues of the complex to that of the crystal structure (PDB code: 1grn), based on the calculation using Drugscore[PPI] [31]. Interface residue was scored based on the binding free energy and degree of burial of each residue forming the RhoA interface. The predicted structures of GAP domain with the N- and C-terminal loop regions (residue number: 658–898) or N-terminal C1-loop domain (residue number: 611–886) were superimposed onto the crystallized RhoGAP-RhoA complexes (PDB: 1ow3, 3msx, and 5irc). The interface residues calculated from DrugScore[PPI] software were mapped on the alignment of the PARG1 GAP sequences with p50RhoGAP (1ow3), ArhGAP20 (3msx), and p190RhoGAP (5irc) (S2 Fig). The theoretical interaction interface was also checked with modeled structure based on proximity of conserved regions of GAP domain to RhoA substrate (S4 Fig). The protein-protein interactions were analyzed and visualized using Chimera software version 1.15.

## Molecular dynamics (MD) simulations

Selected WT PARG1 C1-GAP domain and RhoA-bound structures of the GAP domains with C1-linker region of WT PARG1, T622M, and I845V mutants were subjected to MD simulations using Desmond Molecular Dynamics System, version 5.2 (D. E. Shaw Research, New York, NY) [32]. First, the structure was preprocessed with Protein preparation Wizard of Maestro (version 11.4), the GUI for Desmond, to assign bond orders, add hydrogens, and create disulfide bonds. Then, it was solvated in a box with a buffer distance of 10 Å to the boundary. Afterward, solvation was performed in a box with a buffer distance to the boundary of 10 Å. Sodium and chloride ions were added to neutralize the entire solvated system. Amber99SB-ILDN force field [33] and TIP3P model [34] were used for the protein and water molecules, respectively. After relaxing the system according to the Maestro's default relaxation protocol, an MD run was performed in the constant-NPT ensemble at 310K and 1.0 bar for 2 μs. The coordinates were recorded every 2 ns to yield 1001 snapshots. Otherwise, the same parameter setting as previously reported [35] was adopted. The simulation of WT PARG1 C1-GAP domain alone was performed for 10ns. The protein structure and protein-protein interactions were visualized using Chimera software version 1.15.

## MD trajectory analysis

The resulting MD trajectory was analyzed in the same manner as reported previously [35]. First, it was equidistantly divided into 101 frames so that each frame could contain ten consecutive snapshots. Then the Root Mean Square Deviation (RMSD) values between main chains of arbitrary two frames were calculated to generate a RMSD matrix. In the matrix, frames with a RMSD less than 2 Å were assigned to belong to the same cluster. In each cluster, the frame with the minimal RMSDs to the other members was considered a representative structure of the cluster.

The root mean square fluctuation (RMSF) is a measure of the fluctuation of an atom around its average position. The RMSF of the C-alpha carbon at the protein backbone was calculated from the MD snapshots from trajectory of the 2-μs MD simulation using Bio3d software. Residue interaction analysis for identification of interface residues of wild-type and mutant PARG1-RhoA complexes were conducted by hydrogen bonding analysis using Visual Molecular Dynamics (VMD) [36]. Salt-bridge analysis was performed using VMD setting a distance cutoff 3.2Å between the oxygen of an acidic

residue and the nitrogen of a basic residue. Principal component analysis (PCA) was performed from MD data from every 20-ns in 2-μs MD simulation using VMD and Bio3D package [36–38]. Ramachandran plot analysis was performed based on indicated periods (Figs 6a, S6, S8 and S9) from 2-μs simulation. The trajectory data was retrieved from the structural data in the periods that a stable complex maintained during 100-ns MD simulation for AlphaFold-modeled complexes or entire 2-μs simulation period. Overall motions of the C1-GAP domain interacting with RhoA was varied through complex formation. Major motion of the RhoA substrate along specific directions indicated by eigenvectors was retrieved from the trajectory of PARG1 complex and indicated based on normal mode analysis (S10 Fig) [37,38]. Cross-correlation matrix indicated correlated or anti-correlated motions of RhoA protein bound to the wild-type or mutant C1-GAP domain (Fig 8b) [37,38]. The binding free energies of each stable complex during 2-μs MD simulation were calculated from the representative trajectories showing minimal RMSD in the largest cluster that was categorized by 2D-RMSD. The analysis was performed from the snapshots for 1 frame, using gmx_MMPBSA embedded in GROMACS system (Table 1) [39,40]. This analysis was based on the difference of free energy of the complex, the C1-GAP (receptor), and RhoA (ligand) using the equation, i.e., ΔG binding free energy = G complex − (ΔG receptor + ΔG ligand) [40]. The selection of representative trajectory was performed and used as input parameter file of MM-PBSA analysis. Alanine scanning analysis based on MM-PBSA studies was performed for WT PARG1 to examine the mutation effect on Ile[845] residue in the GAP domain [40].

## Results

### The modeled structure of PARG1 RhoGAP with N-terminal C1 domain showed RhoA specificity

PARG1 contains the C1 domain adjacent to the RhoGAP (GAP) domain (Fig 1a), and several mutations, including missense, nonsense, and frameshift mutations in its entire region, cause NSCL/P, a common birth defect; however, the underlying molecular mechanisms and PARG function in mammalian development remain poorly understood. To assess the substrate recognition mechanism of the catalytic domain of PARG1, the GAP domain of *Homo sapiens* (UniProt No: Q52LW3–1) was retrieved from InterPro and structurally modeled using SWISS-MODEL based on HHPred-predicted secondary structures (S1 Table) [16,17]. Structural assessment using Ramachandran Plot, ERRAT, and ProSA software selected the model with N-chimerin as the best template (PDB code: 3cxl), with a QMEAN score of 0.69 (Fig 1b) [16–18]. To determine its complex structure with RhoGTPase substrates, docking analysis was performed using HDOCK in template-free mode [26]. The HDOCK program, using the template-free docking mode, theoretically predicted the experimentally validated p50GAP complex with interface residues for the substrate well-matched to the crystal structure (PDB code: 1grn) [30,31]. Docking analysis using the HDOCK and DrugScore[PPI] programs revealed no RhoGTPase-binding model. Therefore, the GAP domain with extended N- and C-terminal flexible regions used for GAP assay (residue number: 658–898) was structurally modeled based on HHPred and trRosetta softwares [3,17,18]. Structural assessment selected the model4 as the best tertiary structure (Fig 1c, S2 Table). HDOCK analysis of the domain selected RhoA (PDB: 5irc) as the substrate in a properly matched interface using Drugscore[PPI] (Figs 1d and S1). The interface residues

**Table 1. Theoretical binding free energies of the predicted C1-GAP structures for RhoA.**

| PARG1-RhoA complex | Theoretical ΔG (kcal/mol) |
| --- | --- |
| WT (C1-GAP domain) | −95.59 |
| p.Ile845Val | −97.91 |
| p.Thr622Met | −77.20 |

Theoretical binding free energies of representative frames in the cluster based on the difference of 2D-RMSD were calculated using gmx_MMPBSA [40]. From simulation trajectories in 2-μs MD simulation, representative trajectories as 1 frame retrieved from 100 frame were used as input in MM-PBSA studies at 310K. The selection of the trajectories was considered from 2D-RMSD analysis of the simulation trajectories at an equal time interval (see Fig 4). The following frames were used for the calculation of binding free energies: WT PARG1 (0.78 μs); p.Ile845Val: (0.94 μs); p.Thr622Met (0.78 μs) based on RMSD values.

of the modeled PARG1 GAP structure were theoretically conserved with those of p190ARhoGAP (PDB ID: 5irc) for RhoA binding, as seen in significant levels of binding free energies of the residues (Figs 1d, S1 and S2). Since the GAP domain with the N- and C-terminal extended loop was predicted to form a tertiary RhoA-bound complex, the highly conserved C1 and GAP regions of human PARG1 (residue number: 611–886) were chosen and modeled to obtain further insights into the substrate recognition mechanism. A PSIPRED search of the corresponding C1 region of PARG1 predicted tandem β-sheet and β-sheet/α-helix units connected by loop structure, and the secondary structure matched well with the selected model2 tertiary structure (Fig 1e, S1 and S2 Tables) [17,19,41]. 10-ns MD simulation was performed, and the root mean square deviation (RMSD) plot indicated that the predicted RhoGAP domain was stable (S3a Fig). The C-terminal linker region affected the flexibility of Val$^{676}$ in the $\alpha_1$ helix and Tyr$^{746}$ and Gln$^{749}$ that locate in $\alpha_4$ helix in the positively-charged interface (Fig 1e). The RMSF evaluation revealed that the 20 N- and C-terminal residues of the C1 domain were more flexible than those of the RhoGAP domain (S3b Fig). PARG1 has a separate flexible C1 region and a catalytic interface in the RhoGAP domain (S3b Fig). HDOCK analysis of the C1 domain-containing RhoGAP selected RhoA as the substrate in a properly matched interface with conserved residues with a high theoretical affinity (Figs 1f, S2 and S4) [2]. Although RhoC is inactivated in VE-cadherin-mediated cell–cell contact during cyclic AMP-induced ArhGAP29 activation [42], RhoC and other RhoGTPases, including RhoB, RhoD, RhoG, Rac1, Cdc42, and Tc10, were disregarded as RhoGAP substrates using HDOCK in template-free mode. The arginine finger and conserved interface residues Arg$^{748}$(positively-charged interface), Asn$^{841}$ and Ile$^{845}$ (GAP domain C-terminal interface) played critical roles in RhoA recognition (Fig 1g). These structural positions of interface residues for RhoA binding were conserved with those of crystallized RhoA complexes bound to p50RhoGAP (PDB: 1ow3, ArhGAP20 (PDB: 3msx), and p190ARhoGAP (PDB: 5irc) (Figs 2 and S2). The calculation using Drugscore$^{PPI}$ showed that predicted interface residues of PARG1 GAP domain containing the C1-domain were well-matched to those of the other RhoGAP complexes bound to RhoA, compared to the RhoA-bound GAP domain with the N-and C-terminal regions (S1 Fig).

The C1-GAP domain was also structurally modeled using AlphaFold software [43]. The predicted structure was subjected to HDOCK docking analysis with template-free mode, using crystallized structures of RhoGTPases, and Cdc42 but not RhoA was selected as its substrate. The corresponding C1-GAP region of wild-type PARG1 bound to RhoA (residue number: 2−185) was also structurally modeled using AlphaFold multimer, and model1 was selected based on the overall RMSD scores of backbones between RhoA-aligned C1-GAP domains and 1ow3, 3msx or 5irc. Model3 was also selected by Cα RMSD values between RhoA-aligned C1-GAP domains and 5c2k among five candidate complexes. The selected models possessed interface residues well-matched to the crystallized RhoGAP complexes (Figs 1h, 1i and S5). Compared to modeled PARG1 structure bound to RhoA selected by HDOCK program (Fig 1e and 1f), these complexes showed acceptable levels of DockQ score in model 1 and 3 for 0.348 and 0.345, respectively (acceptable quality: 0.23–0.49).

Several mutations in NSCL/P have been detected in PARG1, which manifest as common birth defects at a frequency of approximately one in 700 births [4,7] (https://www.hgmd.cf.ac.uk/ac/index.php). The C1-GAP domains containing missense mutations p.Thr622Met and p.Ile845Val were structurally modeled using trRosetta and HHPred programs (residue number: 611–886), and superimposed onto the deduced WT C1-GAP structure (Fig 3a). The Thr$^{622}$ residue was predicted to form a loop region, whereas the mutated Arg$^{616}$ residue was located in β-sheet in the C1 domain (Fig 3b). In this study, Arg$^{616}$ and Thr$^{622}$ residues were located within the flexible region of the C1 domain, and a point mutation at Thr$^{622}$ affected a more flexible region than the Arg$^{616}$ residue in the C1-GAP domain during the 10-ns MD simulation (S3c Fig). The mutated Ile$^{845}$ residue is located in the $\alpha_9$ helix of GAP domain (Fig 3a and S3c). Ile$^{845}$ residue of PARG1 was predicted to form RhoA interface and structurally matched to the interface Val$^{198}$ and Cys$^{512}$ residues of p50RhoGAP and ArhGAP20 complexes bound to RhoA, respectively (Figs 2 and S2). HDOCK analysis of the T622M and I845V mutants selected RhoA (PDB ID: 5c2k) as their substrate, and the side chains of the predicted interaction interface residues showed similar orientations between each mutant GAP domain and WT PARG1 (Figs 1f, 1g, 3c and 3d). The binding free energies and

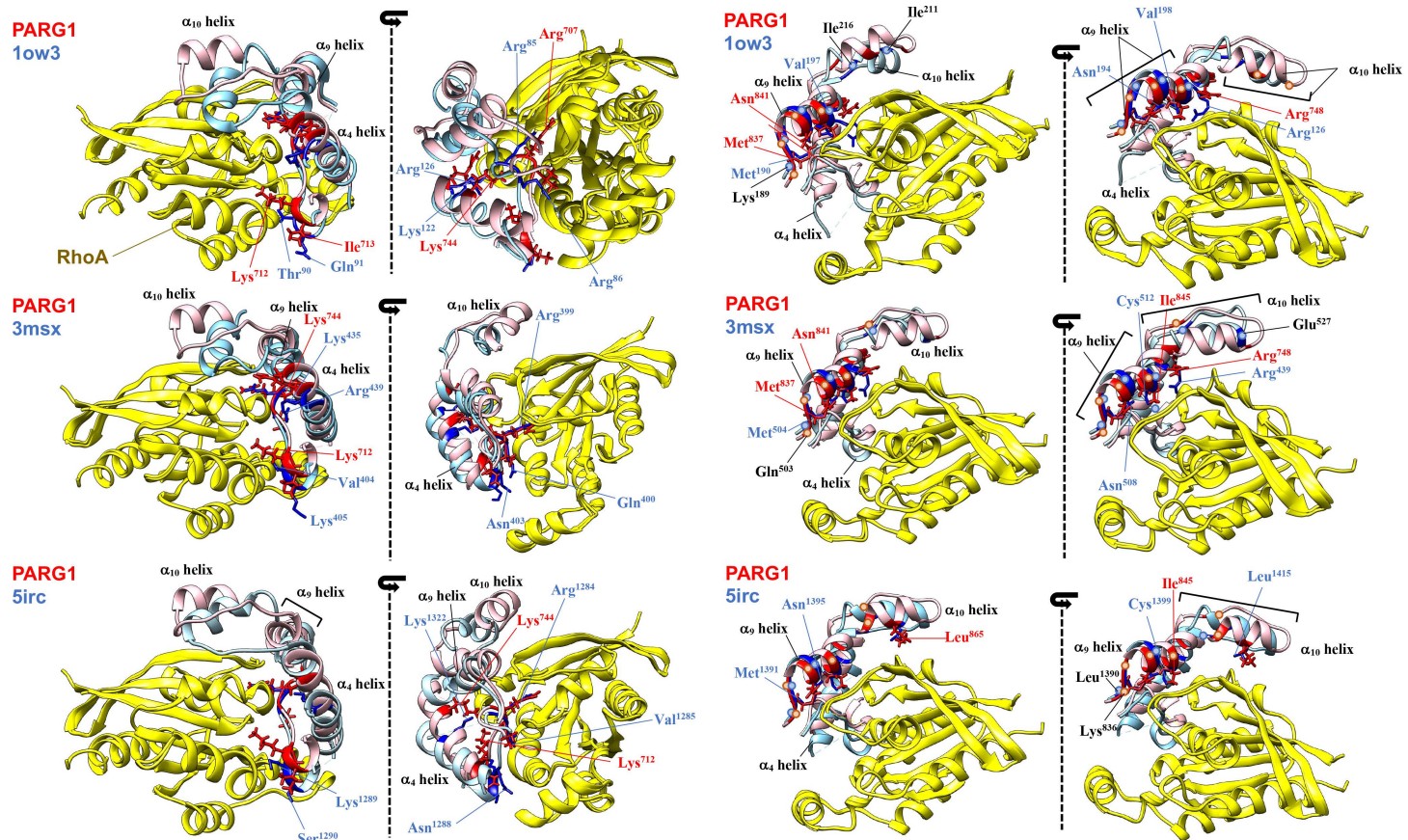

**Fig 2. Superposition of conserved PARG1 C1-RhoGAP domains bound to RhoA protein.** PARG1 GAP domain possesses the conserved regions consist of catalytic arginine finger, positively-charged residues in $\alpha_4$ helix (left figures), and the C-terminal GAP interface to RhoA (right figures) (Figs S2 and S4). The conserved regions of PARG1 C1-RhoGAP domain (pink) face to RhoA (yellow) are superimposed to the corresponding regions of crystallized RhoGAP-RhoA complexes (light blue, PDB ID: 1ow3, 3msx, and 5irc). The side-chains of the interface residues selected using Drugscore[PPI] are indicated on the cartoon (red: PARG1; blue: p50RhoGAP, ArhGAP20, or p190ARhoGAP), and the critical residues of PARG1 obtained by Ramachandran plot of MD simulation in GAP structure (see Fig 6a and red circles in S2 Fig) are marked by red letter with the corresponding residues (blue letter) in the crystallized RhoGAP structure. Side chains of the conserved positively-charged residues and C-terminal interface residues showed similar orientation on the tertiary helical structure to the corresponding residues of each compared RhoGAP structure interacting with RhoA.

degree of buriedness of these mutant C1-GAP complexes with RhoA showed patterns similar to those of WT PARG1 complex by interface analysis using the Drugscore[PPI] software (S1 Fig).

## MD simulation revealed the C1 domain of WT PARG1 RhoGAP formed RhoA interface but not the C1 domains of mutant PARG1

In order to obtain insights into conformational changes in the C1-GAP region, 2-μs MD simulation of WT PARG1 and two missense mutants that interact with RhoA was conducted using Desmond Molecular Dynamics System, version 5.2, based on selected structural models using the HDOCK program [32]. All the structural models showed stable interaction with RhoA throughout the simulation periods (Fig 4a–4c). Interestingly, WT PARG1 formed RhoA-bound conformations through C1-loop region: the N-terminal C1-linker region formed interfaces for the RhoA substrate in addition to its GAP domain in 2-μs MD simulation. However, the predicted structure of I845V and T622M missense mutants showed distinct motion dynamics in its C1 domain, compared to those of WT PARG1 that interacted with the $\alpha_3$-helix interface residues of

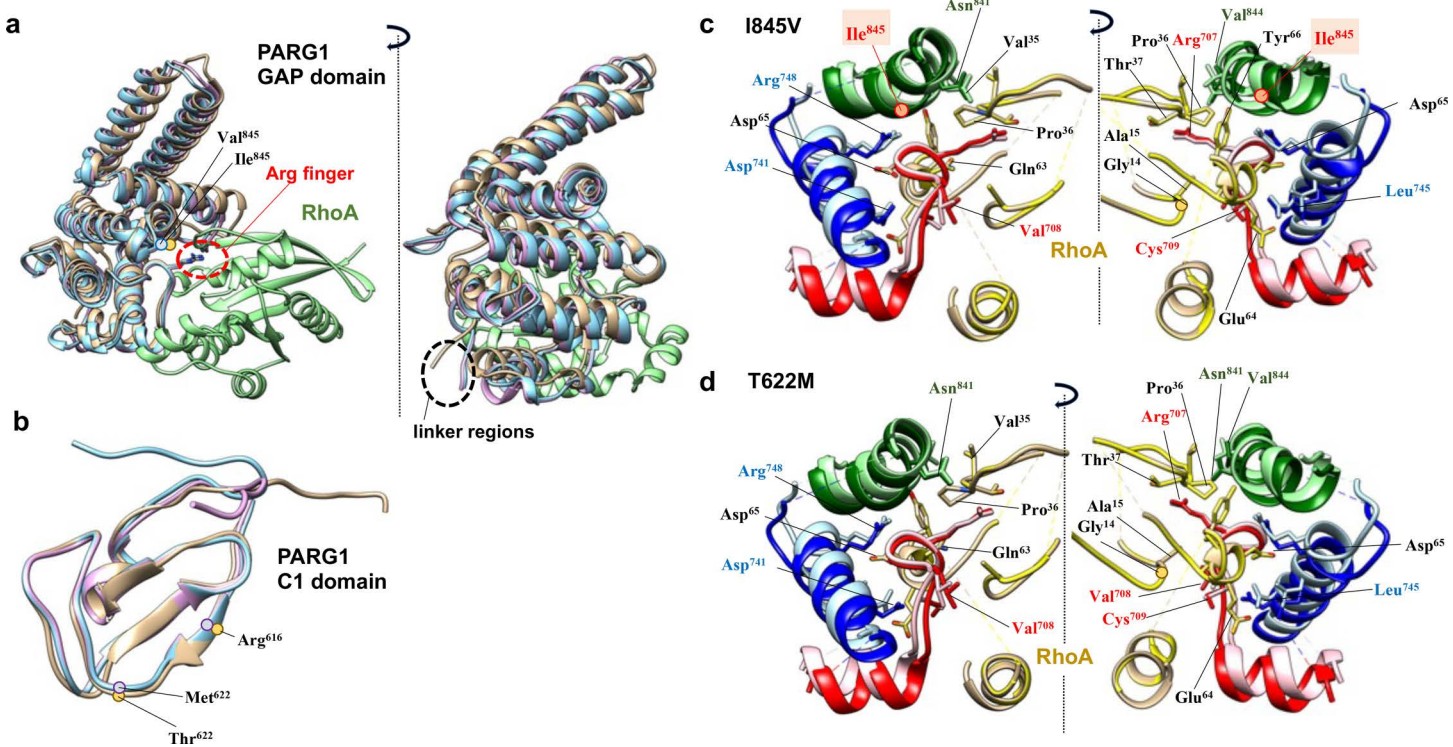

**Fig 3. Modeled structures of C1-RhoGAP domain of WT PARG1, I845V and T622M mutants.** Superposition of (**a**) the GAP and (**b**) C1 domains of modeled complexes of WT PARG1 (residue number: 611-886), and I845V and T622M mutants with RhoA proteins. (**a**) WT PARG1 GAP domain (gold) complexed with RhoA (green) was superimposed to RhoA-bound I845V (light blue) and T622M mutants (purple). (**b**) The C1 domain structure of WT PARG1 was also superimposed onto that of I845V or T622M mutants. The positions of the mutated residues are mapped on the predicted domain structures. (**c, d**) The side chains of mutant PARG1 RhoGAP interfaces for RhoA (< 5 Å in distance; red: catalytic loop; blue: positively-charged interface; dark green: C-terminal interface; yellow: RhoA), and the switch regions and p-loop (black letters) of RhoA were superimposed to the predicted GAP catalytic domain of WT PARG1 (corresponding light-colored ribbon).

RhoA (upper left panel in Fig 5a, RhoA-bound complex structure at 0.78 µs). I845V mutant complex possessed interface residues to the C-terminal $\alpha_4$-helix (Met[134]-Glu[137] and Glu[143] residues) of RhoA and motion of the RhoA was highly dynamic, as seen in hydrogen bond interactions of catalytic Arg[707] residues with Pro[31]-Val[33], Val[35], Thr[37], Asp[59], and Gln[63] residues of RhoA (Fig 5a, lower left panel). In contrast, T622M mutant interacted with more restricted region as Lys[98] residue in the $\alpha_3$ helix of RhoA protein (Fig 5a, right panel). The interface Lys[614,] Arg[616], Lys[617] and Arg[619] residues of WT PARG1 for RhoA locates in the N-terminal loop in the predicted C1 domain structure, and the C-terminal Leu[653]-His[659] residues also formed H-H bonds with the adjacent C1 residues or Arg[748] residue of the GAP domain (Fig 5a). Several conserved residues in the GAP region, including the catalytic loop (Arg[707] and Lys[712]), the $\alpha_4$ helix (Lys[744] and Arg[748]), and its C-terminal interface region (Lys[836], Asn[838], Lys[840], Asn[841], and Tyr[868]), stably interacted with the p-loop and switch I and II regions of the RhoA protein (upper left panel in Fig 5a). The Cα backbone of RhoA-bound WT PARG1 showed stable dynamics at an average of 9.23Å (Fig 4a) that was higher RMSD value than those calculated in I845V and T622 mutants with RMSD of 8.19Å and 6.37Å, respectively (Fig 4b and 4c). 2D-RMSD data revealed that WT PARG1 showed several stable clusters within 2Å through 2-µs MD simulation and the structural model at 0.78 µs was selected as the stable representative conformation (Fig 4d). In contrast, the trajectories of T622M mutant complex showed less clusters and the representative structure of I845V mutant complex in the largest central cluster was unstable thermodynamically, compared to that of WT PARG1 complex (Fig 4e and 4f).

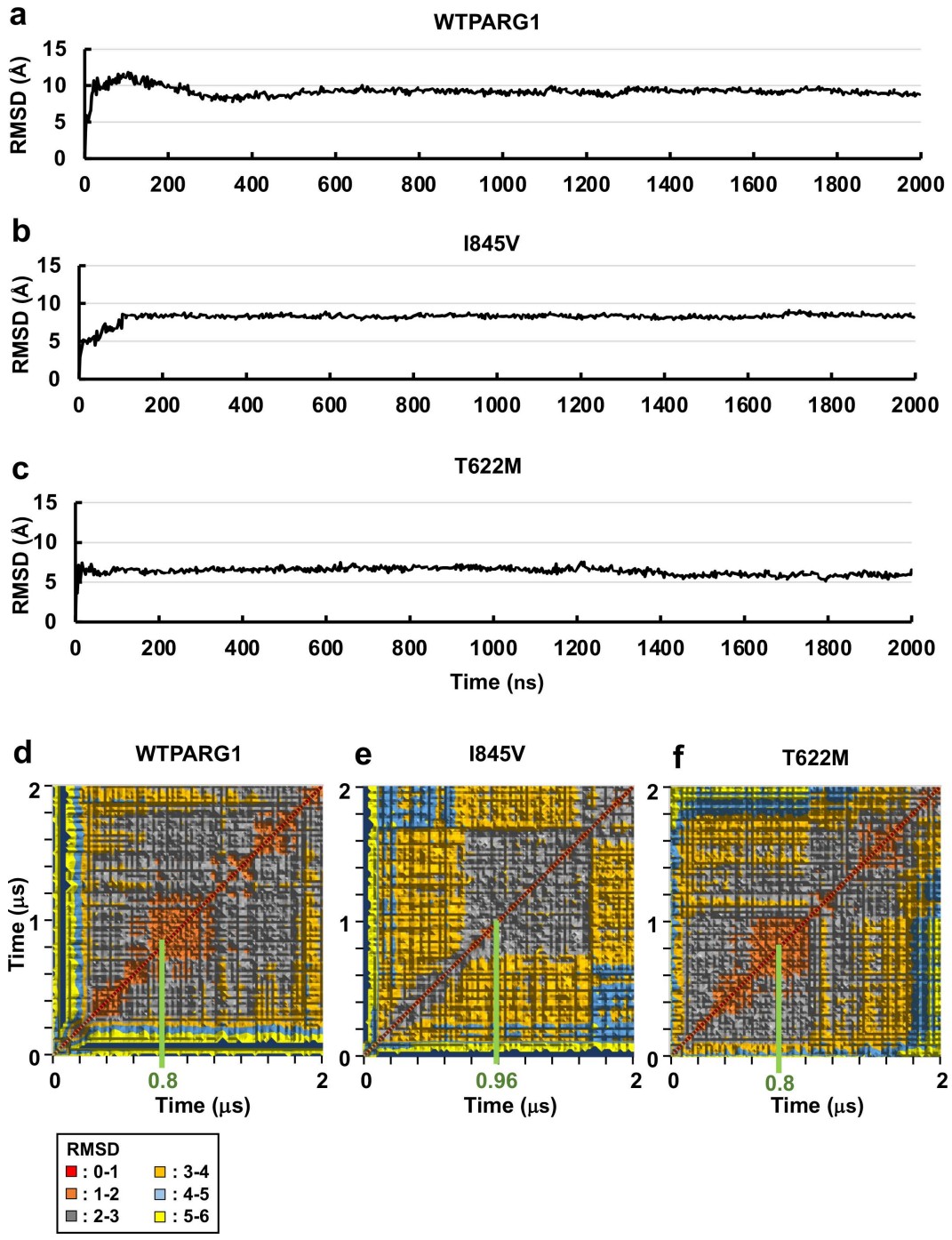

**Fig 4. RMSD graphs for the backbone of WT PARG1, I845V and T622M mutants.** RMSD plots of WT PARG1 (residue number: 611-886) (**a**) and I845V (**b**) or T622M (**c**) mutant for 2-μs MD simulation. The C1-linker formed stable RhoA binding interfaces (**a, b**) in accordance with the GAP region (see Fig 5). Several positively charged residues in the C1 region such as Lys[614] and Arg[619] residues interacted with the RhoA substrate. 2D-RMSD matrix of MD trajectories of the RhoA complexes interacting with WTPARG1 (**d**), I845V (**e**), and T622M (**f**) mutants. The RMSDs of each complex model were calculated and classified into several clusters based on the differences. Among the frames in the clusters, representative WT and T622M complexes were selected at 0.78 μs and the corresponding I845V complex was selected at 0.94μs, respectively, based on the minimal RMSD value within the frames in the largest central cluster.

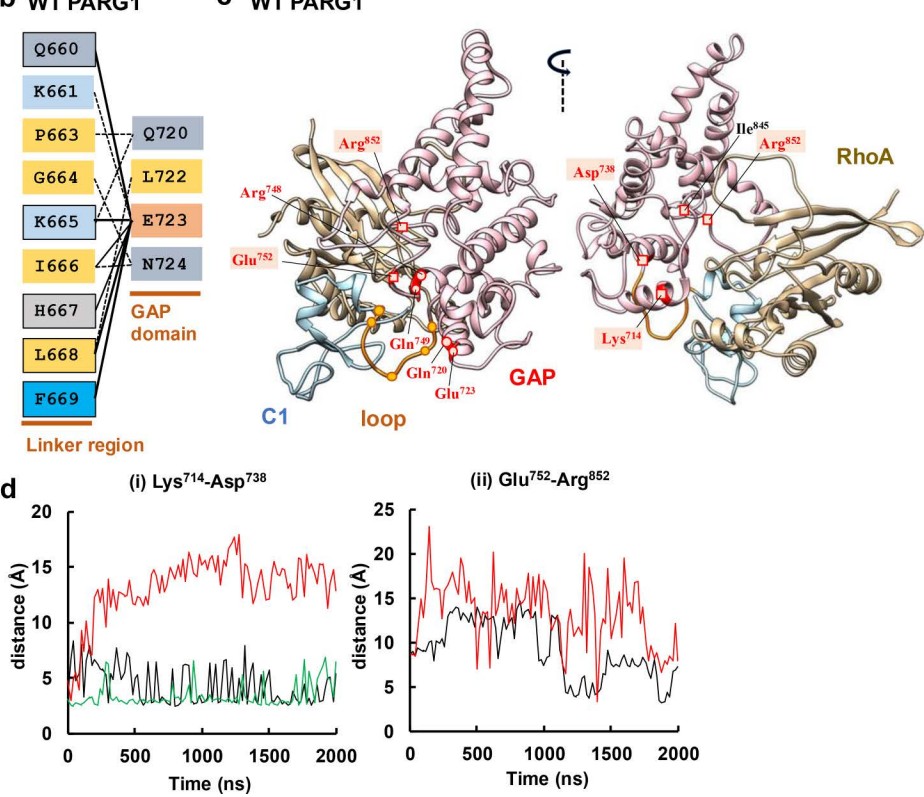

**Fig 5. Catalytic fold of WT PARG1 C1-GAP domain for RhoA interaction.** (a) The motions of the GAP domain with C1-linker of WT PARG1 are indicated (C1 region: light blue; GAP domain: pink; linker region: orange; RhoA substrate: green). The cartoon represented the C1 domain formed the

interface for RhoA with conserved residues of GAP region. The conserved RhoA interface residues within GAP domain are marked with red ribbon and color circles. (Left upper panel). In WT PARG1-RhoA complex, the interface residues in the C1 domain forming hydrogen bonds were formed by its N-terminal residues (Lys[614], Arg[616], Lys[617], and Arg[619] marked by blue) and the interacting residues of RhoA (Glu[93] and Glu[97], marked with blue circle) are indicated. The representative structure of WT PARG1-RhoA complex at 0.78 µs is indicated (C1: blue, loop: orange, GAP: pink, RhoA: green, switch I and II: dark green). The positions of variable residue adjacent to arginine loop, positively-charged residues in $\alpha_4$-helix, and C-terminal interface residues that formed stable hydrogen-bonds with RhoA substrate in 2 µs MD (more than 5%, *i.e.,* 100 ns, in the simulation,) were mapped on the representative structure at 0.78 µs. (Lower left panel) Representative structure of I845V complex at 0.94 µs. The C1 residues of I845V formed interfaces between $\alpha_4$-helix of RhoA. It is noticeable that variable residues adjacent to the arginine catalytic loop formed irregular interfaces to p-loop and switch I of RhoA (see S7 Fig). Irregular torsion of GAP structures was significantly detected in I845V complex. The $\alpha_3$-helix and the C-terminal C1 region showed I845V mutant-specific motions (dotted oval). (Right panel) The representative structure of T622M mutant complex at 0.78 µs. The positions of interface residues in the mutant C1-RhoGAP for RhoA was mapped (more than 5% in the simulation period) and Leu[865] in the C-terminal interface and Asn[711]-Ile[713] residues formed stable hydrogen-bonds with RhoA that was not matched to the contact positions seen in crystallized complexes (see (2), S7 Fig). The C1 region (residue number: 610-659) of T622M mutant was superimposed to the corresponding region of WT PARG1 and specific residues showing distinct motions between the complexes are indicated (S6 Fig). (**b**) Predicted interactions of the linker region and the residues of the $\alpha_3$-helix and $\alpha_3$-$\alpha_4$ loop of WT PARG1 GAP domain in the catalytic complex. Bold line indicates the stable hydrogen bonding detected more than 100 ns in the WT PARG1 trajectory ensembles. Thin lines and dotted lines indicate hydrogen bond interactions detected more than 20-ns and 2 ns, respectively, in 2-µs trajectory of WT PARG1. (**c**) The interacting residues of the C1-linker region and the $\alpha_3$-helix with its C-terminal loop were mapped on the WT PARG1-RhoA complex. The position of Lys[714] and Asp[738] residues in the $\alpha_3$ and $\alpha_4$ helices, Glu[752] residue in the $\alpha_4$-$\alpha_5$, and the Arg[852] residue in the $\alpha_9$-$\alpha_{10}$ loop was mapped on the cartoon as red rectangles in the WT PARG1 GAP domain. Interactions of Gln[660], Lys[665], Ile[667]-Phe[669] residues in the linker region (orange) and Glu[723] residues in the C-terminal $\alpha_3$-helix affected the conformation of GAP domain distinct from the C1 and GAP interfaces for RhoA protein. The positions are indicated based on WT PARG1 complex structure at 0.78-µs period. (**d**) The changes in residue-residue distances forming salt-bridges in the GAP fold in the RhoA complexes. (**i**) Lys[714]-Asp[738] and (**ii**) Glu[752]-Arg[852] residue pairs were detected in the WT PARG1 (black line), I845V (red line), and T622M mutant complexes (green line) in 2µs MD simulations. Based on the periodical fluctuating features of the conserved regions detected in the residue-residue distances as 65-70 ns, 100 ns was selected as the threshold for the stable interaction.

Secondary structures of the C1 domain of WT PARG1 were highly preserved between stable complexes in the representative trajectories of 2-µs simulations, despite distinct motions of Lys[648] in the C1 domain and Cys[657]-Phe[669] residues in the loop region from mutant complexes (Fig 5a, upper right panel, S5 and S6a–S6e Figs). Especially, Cys[649]-His[659] residues in the C-terminal region of the WT PARG1 C1 domain formed coiled-coil structure during stable complex formation, which was in contrast to a stretched loop detected in the corresponding region of I845V complex (Figs 5a lower left panel, 5c and S6). Irregular torsion of GAP structures was detected significantly as the difference in residue-residue distances between the stretched region and Asn[714] residues in the $\alpha_3$, Asn[714] and Asp[738] residues in the $\alpha_4$ (Fig 5d-i), or the $\alpha_9$-$\alpha_{10}$ loop (Arg[852] residue) and $\alpha_4$ helices including Glu[752] residue in I845V complex (Fig 5d-ii), which may be caused by the disorganized interface in the $\alpha_9$ helix (S8 Fig). In the case of T622M mutant complex, mutated methionine caused the steric interference for the motion of apposed β-sheet in the C1 region, although the partial coiled structure was formed in the C-terminal C1 region (S6e–S6i Figs). In WT-specific patterns, the C-terminal linker (residue number: 660–669) interacted with the $\alpha_3$ and $\alpha_3$-$\alpha_4$ loop of the GAP domain and the Glu[723] residue had a central role for stabilization of the loop region (Fig 5c and 5d).

These interface residues of WT PARG1 complexes were well-matched to the conserved interface residues between the predicted PARG1 RhoA complex structure and other crystallized RhoGAP-RhoA complexes (Figs 1f, 1g, S7 and S9). In contrast, the I845V or the T622M mutant contained interface residues that were not matched to the corresponding positions of crystallized RhoGAP-RhoA complexes (S7 Fig). These motion properties were reflected by the binding free energies of the complex calculated using gmx_MMPBSA (Table 1) [40]. The binding free energy of stable I845V was similar to that of WT PARG1-RhoA complex (−95.59 kcal/mol) calculated from selective representative structures based on 2D-RMSD (Fig 4d and 4e, Table 1). In contrast, representative T622M complex showed lower level of binding free energy than that of WT PARG1 complex (80.8%) (Fig 4f, Table 1).

Cys[657] residues in the C-terminal C1 domain and Leu[668] and Phe[669] residues adjacent to the GAP domain showed specific motion characteristics of WT PARG1 in contrast to Thr[622] residue (Figs 6a and S6a–S6d). The flexibility of the Arg[647]-Gly[658] residues detected as a coiled-coil structure may reflect the stability of the RhoA-bound complex, as specific range of

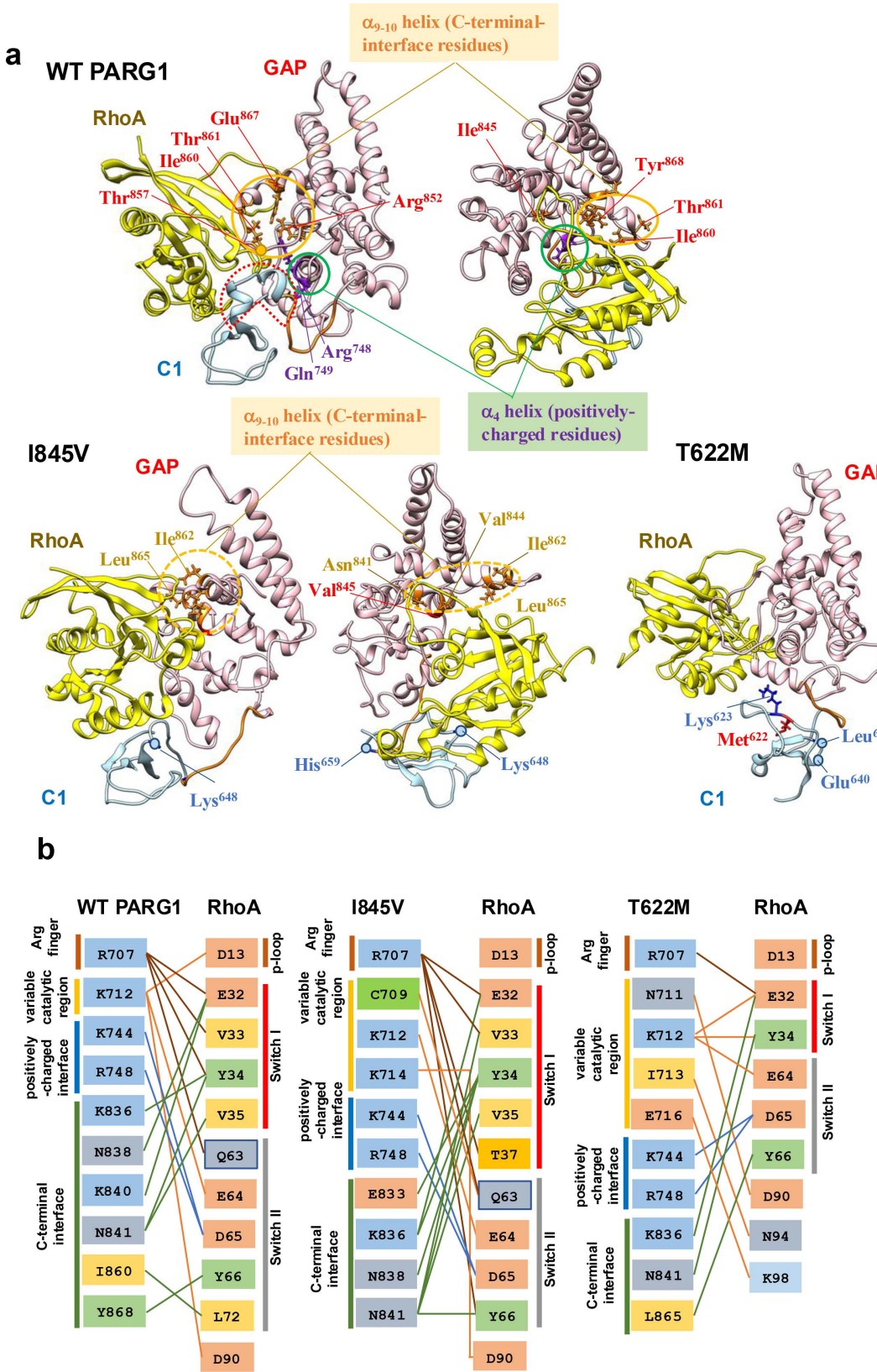

**Fig 6.** Continued.

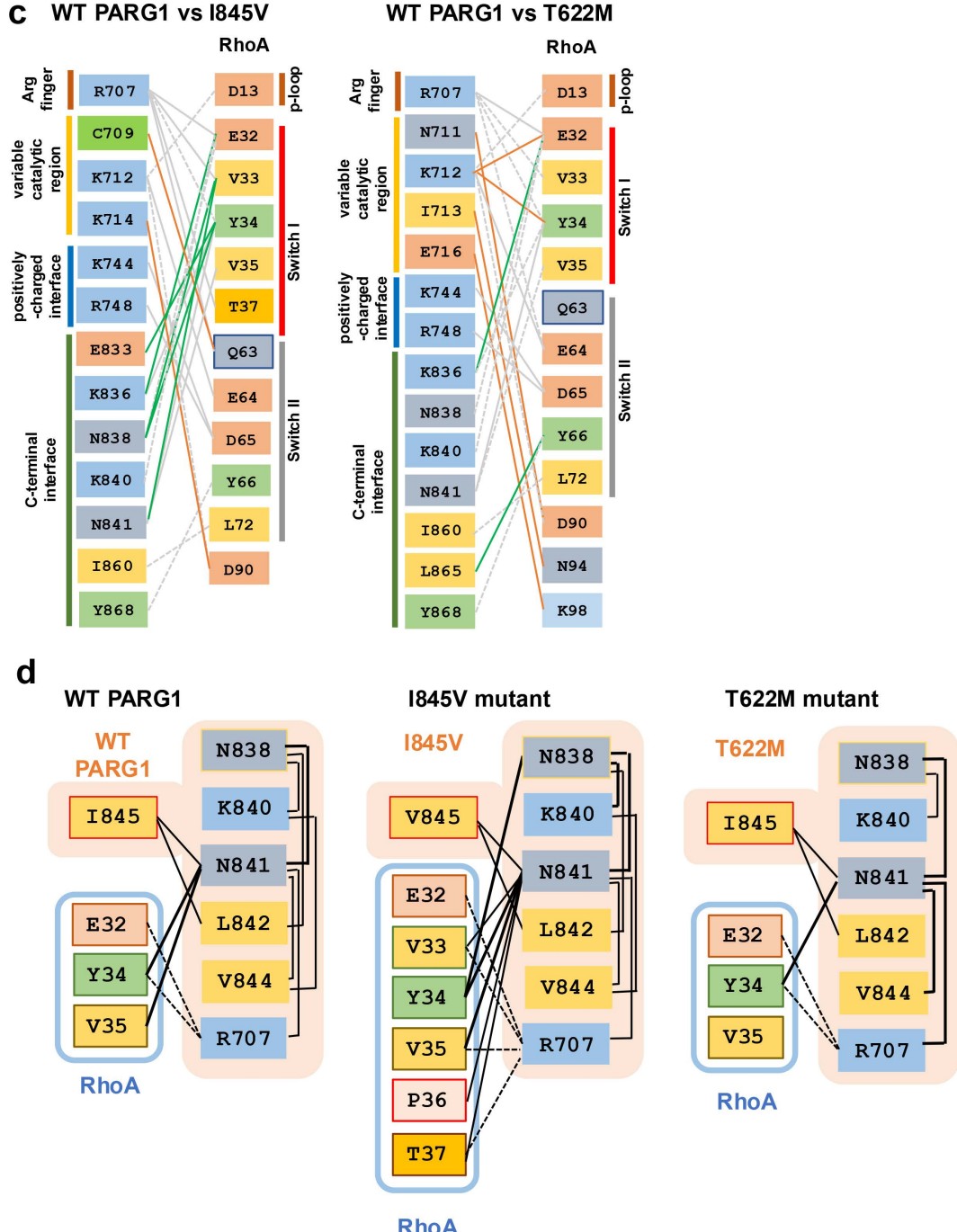

**Fig 6. Residue interaction networks showing the interactions between important residues in RhoA-bound PARG1 complexes.** (a) The critical residues characteristic of WT and mutant RhoGAP-RhoA complex-specific motions. The interface residues located in the C-terminal $\alpha_4$-helix (Arg[748] and Gln[749] residues: purple ribbon), $\alpha_9$-$_{10}$ loop (Arg[852], Thr[857], Ile[860] and Thr[861] residues) and C-terminal region of $\alpha_{10}$-helix (Tyr[868] residue) showing WT-specific motions are presented based on Ramachandran plot analysis of MD simulation of the C1-GAP domains (see Fig. S8 and S9a-d). The C-terminal C1 domain and loop region formed a coiled-coil structure in WT PARG1 complex (red dotted line). Both the Thr[622] and Ile[845] mutations to Met[622] and Val[845], indicated in red letters, showed partial changes in the motion angle, comparing to the counterparts interacting with RhoA (see Fig. S8a and S6f). I845V mutation mainly caused changes in motion angles of Asn[841] and Val[844] residues in the $\alpha_9$-helix and Ile[862] and Leu[865] residues in N-terminal $\alpha_{10}$ helix as shown in dark yellow (Fig.S8b-e). In addition to the interface residues in C-terminal $\alpha_4$-helix (Arg[748] and Gln[749] residues), adjacent Lys[623] or Glu[640] and Leu[642] residues in beta$_{2-3}$ sheets in the C1 region (indicated in blue ribbon) showed different motion angles in T622M complex. (b) The hydrogen bond

interactions of the conserved residues in the arginine finger, positively charged and C-terminal interfaces of WT PARG1, I845V, and T622M mutants to RhoA are indicated based on 2μs MD simulation ensembles (arginine finger: brown line; structurally conserved region close to the Arg[707] residue (variable1, corresponding to Fig.S4): orange line, positively charged interface: blue line; C-terminal interface: green line, p-loop; dark yellow line; switch I: red line, switch II: gray line). (**c**) Changes in hydrogen bond network in wild-type versus mutant PARG1 complexes. Mutant-specific hydrogen bonds are highlighted in color lines as shown in (**b**). Conserved hydrogen bond interactions with WT PARG1 complex (dark gray) and WT PARG1 complex-specific hydrogen bonds (light gray, dotted line) were embedded in the interaction map. (**d**) The hydrogen bond interactions of Ile[845] (WT PARG1 and T622M mutant) or mutated Val[845] residue (I845V mutant) with the C-terminal interface residues (thin line: H-bond detected more than cumulative 4ns period (0.2%); bold line: hydrogen bond formed more than cumulative 20ns period (1%) in 2μs MD simulation, respectively) retrieved from MD ensembles. The hydrogen bonds in the catalytic arginine finger (Arg[707]) with the RhoA interface are indicated as dotted lines. The cutoff value of hydrogen bond interaction was selected as 3.0Å more than 4 ns period to properly detect the stable hydrogen bond formation in motion dynamics.

ψ-φ angles in the specific range of torsion of the C-terminal C1 region were detected in the stable complex formed during the simulation. (Figs 6a and S6). A difference in the φ angular motion of the Ile[845] mutation to Val[845] residue in the GAP domain was detected by Ramachandran plot analysis (S8a Fig). Mutation of Ile[845] to Val[845] residue caused changes in motion angles of adjacent Asn[841] and Val[844] residues in the $α_9$-helix and Ile[862] and Leu[865] residues located in $α_{10}$-helix (Figs 6 and S8b–S8e). The salt-bridge distance between the $α_4$ helix (Asp[738] residue) and Lys[714] residues located in the $α_3$-helix or between the $α_4$-$α_5$ loop (Glu[752] residue) and the Arg[852] residue in the $α_9$-$α_{10}$ loop was greater than that of WT PARG1 complex (Fig 5d). Extension of Lys[648]-His[659] loop was characteristic of I845V mutant complex, in contrast to WT PARG1 complex, and distortion of the $α_3$-helix was accompanied to motion of the loop. T622M mutation caused restrictive motions of Glu[640] and Leu[642] residues in loop region between the $β_2$-$β_3$ sheets in the C1 region, due to steric hinderance imposed by sidechain of Met[622] residue (S6f–S6h Fig). Differences in motion angles of the Arg[748] and Gln[749] interface residues in C-terminal $α_4$-helix were detected in T622M complex (S8g and S8h Fig).

The interaction network between each C1-GAP domain was analyzed (Fig 6b and 6c). The interface residues for the RhoA interaction with PARG1 complexes showed distinct patterns depending on the C1-linker motion between WT and mutant RhoA complexes (Fig 6b). The switch I region of the RhoA protein interacts with the catalytic arginine finger (Arg[707]) and C-terminal interface (Lys[836], Asn[838], Lys[840], and Asn[841]) of the WT GAP domain. The C-terminal interface residues in the GAP domain of the I845V and T622M mutants in 2-μs MD simulation tended to form fewer hydrogen bonds with the switch II regions and p-loop of RhoA than with the corresponding regions of the WT complex (Fig 6b and 6c). Furthermore, in I845V mutant, the interface residues in $α_8$-$α_9$ loop and N-terminal $α_9$ helix in the C-terminal interface formed stable hydrogen bonds to the switch I of RhoA (Fig 6b and 6c). In contrast, C-terminal interface of T622M mutant elicited less stable hydrogen bond interactions with the switch I region of the RhoA protein in contrast to the interaction of variable catalytic region with $α_3$-helix of RhoA (Fig 6b). In WT PARG1 complex, the Asn[841] residue in the C-terminal interface formed stable hydrogen bonds with the Ile[845] residue and Tyr[34] and Val[35] residues in the switch I region of RhoA, suggesting that theoretically conserved GAP interfaces properly recognize the RhoA protein through cooperative motions of the C1-linker domain (Figs 6d and S7). The corresponding Val[845] residue of the mutant C1-GAP domains formed a hydrogen bond with the conserved Asn[841] residue of the C-terminal interface which is corresponding to the Asn residue required for stable RhoA binding in the crystallized RhoGAP complexes in 2-μs MD simulations (Fig 6d). However, Asn[841] residue in the I845V mutant formed wider range of conserved residues in the switch I region, and the corresponding interface residue of T622M mutant formed less hydrogen bonds to Tyr[34] of RhoA and $α_9$ interface residues than that of WT complex (Figs 6d and S7). This MD simulation analysis suggests that mutated residues in both the C1 and GAP domains affect the catalytic folding of the GAP domain on the RhoA substrate.

The modeled C1-GAP regions of wild-type PARG1 bound to RhoA (model 1 and 3) using AlphaFold multimer (Fig 1h and 1i) were also analyzed for MD simulation. The selected complex model was unstable through MD simulation with irreversible mode, and no interaction of the C1 domain with the RhoA substrate was detected in both cases (data not shown). The binding free energy of its stable RhoA complex as model 3 calculated using MM-PBSA was −60.27 kcal/mol, suggesting that the C1 loop region is required for stable association of PARG1 with the RhoA substrate.

## The regional motions of the predicted C1-GAP structures of WT and mutant PARG1 bound to RhoA proteins

Domain regions displaying higher levels of flexibility were analyzed, and the backbone of the C1-loop domain of I845V was found to be less flexible at residues 611–660, in contrast to coordinated motions of the region of the WT C1 domain interacting with the RhoA substrate (Fig 7a-i and 7a-ii). The corresponding region of the T622M mutant of the C1-GAP domain showed high levels of flexibility partially maintaining its secondary structures and coiled structure in the C-terminal C1 region, suggesting that the conformation of C1 containing mutated residue contributed to unstable motion dynamics (Fig 7a-iii). Coordinated motions of the C1 region with the GAP domain structure was also detected in WT PARG1 complex in 2-μs trajectory, despite a highly flexible C1 region in contrast to I845V mutant (Figs 5a and 7). In contrast, the Gln$^{675}$ residue embedded in the N-terminal GAP fold showed similar restricted distribution between WT and T622M mutant but not I845V complex. The N-terminal motion of the C1 domain might cause weak RhoA association through the loop region and the $\alpha_3$ helix of GAP region of T622M mutant. The relative RMSF values of the N- and C-terminal C1 regions of WT PARG1 that formed RhoA interfaces were similar levels, and the $\alpha_4$ and $\alpha_7$ helices of the GAP domain were highly stable (Fig 7b-i). In contrast, the relative RMSF scores of the mutant C1 regions were imbalanced between the N- and C-terminal regions (*i.e.,* the N- and C-terminal C1 regions of I845V were less flexible and several regions of T622M mutant were highly flexible, respectively.) (Fig 7b-ii and 7b-iii).

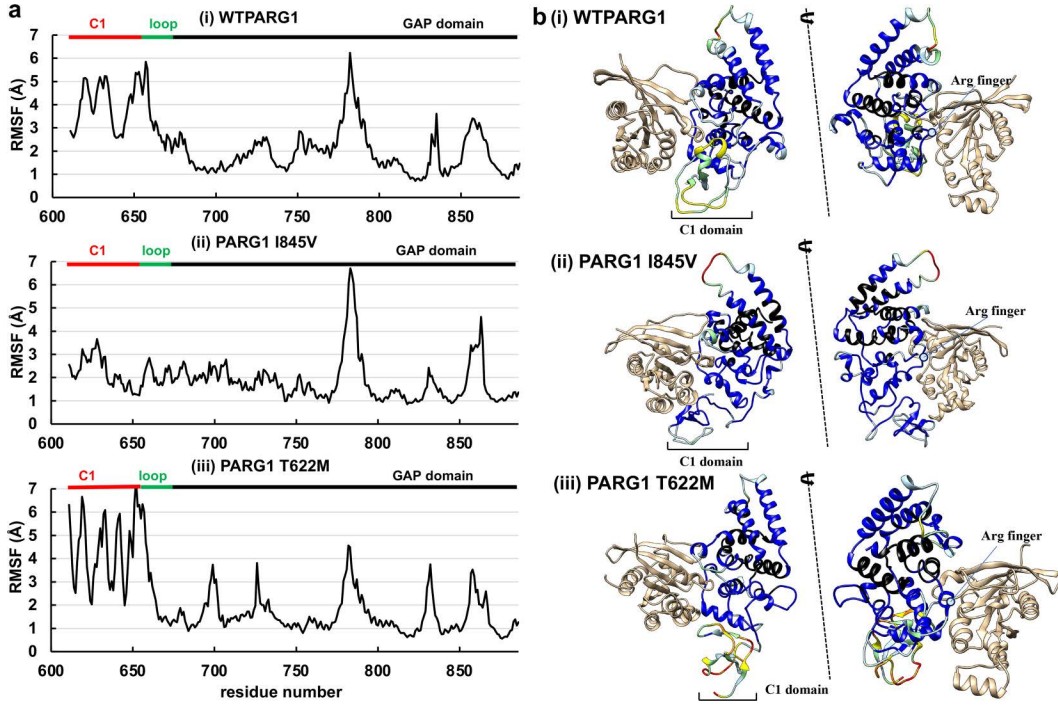

**Fig 7. Motion of C1-GAP domain of WT and mutant RhoGAPs interacting with RhoA substrate. (a)** RMSF values for the GAP domain containing the C1-linker region of PARG1 bound to RhoA in MD simulation. The N-terminal residues of the C1 domain (red: residue number 611–659), linker region (green: residue number 660–670), and the GAP domain (black: residue number 671–886) are indicated with color bars. (**i**) WT PARG1 with the C1-linker interacting with RhoA, (**ii**) I845V PARG1 mutant complex, (**iii**) T622M PARG1 mutant interacting with RhoA in entire 2-μs MD trajectory. **(b)** Structural dynamics of the GAP domains with C1-linker are mapped based on the relative RMSF values compared to the average calculated from each C1-GAP trajectory of the corresponding 2-μs MD simulations in (**a**). The WT PARG1 GAP structure showing interresidue interaction between the GAP and C1 regions (**i**) are mapped based on calculation from 2-μs MD simulation. The flexibility map of stable RhoA-bound complex of PARG1 I845V (**ii**) and T622M mutants (**iii**) through 2-μs period are indicated. The flexibility map was constructed based on the relative RMSF scores: black: <0.5; blue: 0.5~1; light blue: 1~1.5; light green: 1.5~2.0; yellow: 2.0~2.5; orange:2.5~3.0; red:>3.0.

The flexibility of the RhoA protein was also analyzed, and its binding stability to the mutant I845V GAP domain was reflected in its low flexibility in the α$_4$-helix interacting with the C1 domain in 2-μs MD trajectories (Fig 8a-ii). Interestingly, the switch I region (residue number: 31–40) of the RHOA protein was flexible in the stable WT PARG1 complex, which interacted with the conserved residues in the GAP domains (Figs 8 and S7). Low levels of flexibility of the α$_3$-helix were detected in both the WT and T622M mutant complex in which the C1 domains formed hydrogen bonds (Fig 8a-i and 8a-iii). Correlation matrix analysis indicated that more anti-correlated motion between the switch I and β2-β3 loop (residue number: 40–55) was detected in RhoA bound to WT PARG1 than in its mutant counterparts (Fig 8b). Principal component analysis [37,38,44] using 2-μs trajectory data revealed that most of the motion was due to the contribution of the switch I region (residue number: 31–40) for the WT C1-GAP complex, but not for RhoA bound to the mutant C1-GAP regions (Figs 8c, 8d and S10). These results suggested that the C1-linker interfaces of WT PARG1 for RhoA allowed for the proper conformation of conserved patches of its GAP domain and the RhoA protein responsible for recognition, leading to catalytic GTP hydrolysis.

## Discussion

NSCL/P is one of the most common human birth defects with complex traits. In addition to pathogenic variants in genes encoding adhesion components such as the p120-catenin/E-cadherin complex, several syndromic NSCL/P-associated genes, such as PARG1 and IRF6, have been identified [4–7,45,46]. Although genetic etiologies have been recently identified, a biological pathway for NSCL/P has not yet been established. Several risk alleles located in the IRF6 regulatory region are reportedly associated with the penetrance of PARG1 loss-of-function phenotypes in individuals, and both proteins are suggested to be within the same pathway [47–49]. However, several reports have not supported an epistatic interaction between IRF6 SNP and PARG1 [5]. Loss of PARG1 function is associated with syndromic NSCL/P and there is no mutational hotspot in its coding region [7]. The genetic loci of several pathological mutations, including missense or frameshift mutations, have implicated the importance of the C1 and GAP regions in signaling pathways [4,7]. The importance of counter-balanced RhoGTPase signaling in neural crest cells (NCC) and mesenchyme have been indicated in several studies on craniofacial development [50–53]. The development of the lip and palate requires cytoskeletal regulation by several RhoGTPases involved in cellular proliferation, adhesion, and motility. Dominant negative Rho kinase expression under Wnt1-cre in NCC causes severe craniofacial malformation by progressive apoptosis during embryogenesis [50]. Potential role of RhoA and Rho kinase in cell-adhesion to the matrix in the developing craniofacial region are suggested in agreement with its similar abnormalities in Wnt1-cre conditional Rac1 knockout embryos [51,52]. Cell motility in wounded closure was impaired due to enhanced RhoA-GTP and downregulated Cdc42-GTP levels in primary cleft cells from CL/P individuals [53].

PARG1 is ubiquitously expressed in epithelial, mesenchymal, and endothelial cells. Its spatiotemporal expression pattern in the embryo is consistent with its role in craniofacial development [4]. PARG1 possesses higher catalytic activity toward RhoA than toward Rac1 or Cdc42 [3]. Under physiological conditions, the cAMP-mediated Rap1-ArhGAP29 activation pathway inhibits RhoA and RhoC by upregulating endothelial barrier function [42,54]. RhoA is also a substrate of PARG1, which is induced under hypoxic conditions via HIF1 activation [3,55]. Therefore, regarding biological function of PARG1, the substrate recognition mechanism is an important issue. In this study, we computationally modeled the structure of the C1-RhoGAP domain and MD simulation analysis revealed the importance of coordinated motion of the C1 region with GAP domain for stable RhoA recognition. *In silico* structural modeling of the GAP domain containing the N-and C-terminal regions or the adjacent C1 region based on HHPred-based modeling selected the RhoA-bound structures using HDOCK docking analysis in template-free mode. These results were consistent with its monospecificity for RhoA [3,10]. The corresponding region of PARG1 with the C1 domain was also structurally modeled using AlphaFold [43]. Cdc42 (PDB code: 5c2j) was selected as substrates for both AlphaFold-modeled GAP and C1-GAP domains using the HDOCK template-free mode, however the results were not consistent with the previously reported monospecificity of

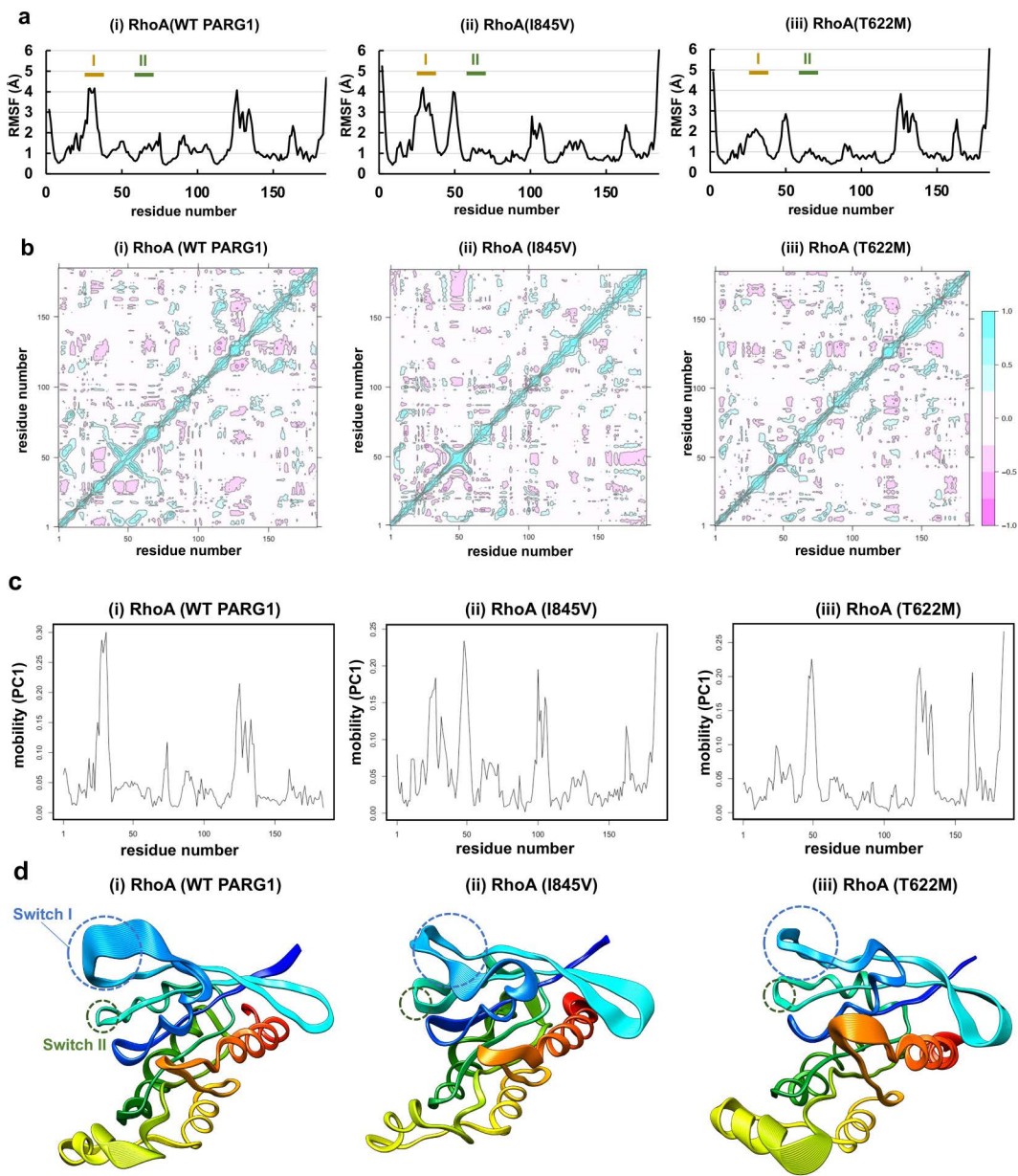

**Fig 8. Motion of RhoA protein bound to the C1-GAP domain of WT or mutant RhoGAPs. (a)** RMSF values for RhoA bound to the GAP domain with C1-linker in MD simulation. The values were calculated from the RhoA ensemble of the stable complex structures with (**i**) WT PARG1, (**ii**) I845V mutant, and (**iii**) T622M mutant during 2-μs MD simulation. The switch I (residue number, 31–40) and switch II (61–70) are indicated with yellow and green bars, respectively. **(b)** The correlation matrix of RhoA bound to (**i**) WT PARG1, (**ii**) I845V, or (**iii**) T622M mutant was calculated based on the trajectory data in (**a**) using Bio3D software. The range of motion is indicated by pink to blue in the panel. Blue indicates positive correlation whereas pink indicates anti-correlation. **(c, d)** Residue-based mobility plot of (**i**) WT PARG1, (**ii**) I845V, or (**iii**) T622M mutant. The eigenvectors and eigenvalues were calculated by diagonalizing the atomic displacement correlation matrix of Cα atom coordinates from RhoA structures. Eigenvalue spectrum is indicated in S10 Fig. The first principal component represents as equidistant atomic displacements from the mean structure. The high values of first principal component in switch I region of RhoA-bound WT C1-GAP region were detected. The RhoA structure was colored from the N-terminus (blue) to C-terminus (red) by UCSF chimera software.

RhoA in PARG1 [3,10]. In addition, AlphaFold multimer C1-GAP models were examined, and the interface residues of the model 1 and 3 with minimal RMSD score between the crystallized structure showed higher theoretical cumulative binding free energies than those with maximal RMSD value. The selected RhoA-bound complexes obtained from AlphaFold multimer possessed similar RhoA interfaces in the GAP region (S5 Fig), and were analyzed for MD simulation, but both models showed an irreversible dissociation of the substrate. The hydrogen bond profiles of the complex models by AlphaFold-multimer showed matched residue pairs to the crystallized complex structures (S11 Fig), but the complexes were not stable. These results suggest the GAP domain alone is not sufficient for stable complex formation in the case of PARG1.

Docking analysis of each crystallized structures of p50, GRAF, and ARAP3 using HDOCK provided theoretical affinities that were comparable to the experimental data [23,56–58]. The selected structure of the PARG1 complex containing the C1-linker showed proper interfaces with RhoA with a high probability of contact obtained by several crystallized structures in view of the residue interaction matrix and the calculated binding free energies, as shown by Amin *et al.* [2] (Figs 2, S2, S4 and S7). Ramachandran plot analysis indicated that mutated Val$^{845}$ residue showed distinct φ angles compared to Ile$^{845}$ residue of WT PARG1, presumably causing torsion of the overall GAP structure (Figs 6a, 9, S8 and S9). Interaction network in MD simulation indicated the C-terminal interface residues in the I845V mutant GAP domain formed more hydrogen bond interactions with switch I residues of RhoA than those of WT and T622M PARG1, which may cause the twisted motions of the RhoA-bound I845V complex through the $\alpha_9$ helix-RhoA interaction (Figs 4c, 6b and S7–S9). Interestingly, the critical residues specific for WT PARG1 dynamics in RhoA recognition such as Arg$^{748}$, Asn$^{841}$, Ile$^{845}$ or Thr$^{857}$-Thr$^{861}$ in $\alpha_9$-$\alpha_{10}$ loops are well-conserved in the predicted interface regions among RhoGAPs bound to RhoA protein (Figs 2 and S2).

Several mutations in the GAP domains have been reported to cause cellular pathologies associated with cytoskeletal dysregulation by promoting or inhibiting their catalytic activities [59]. For examples, Q158R mutation of ArhGAP24 (Fil-GAP) found in focal segmental glomerulosclerosis causes loss of Rac1-GAP activity. Loss of Arhgap24 function specifically expressed in kidney podocytes causes membrane ruffling, leading to the predominance of abnormal Rac1 signaling with kidney damage, despite of hormonal stimulation [59]. In contrast, E313K mutation of α2-chimerin in Duane's retraction syndrome increased GAP activity, which leads to affect Rac1 binding. Enhanced GAP function with reduced Rac-GTP levels, caused by hyperactivated chimerin mutant, results in dysregulation of oculomotor nerve development [60]. Both mutated sites locate near the position of the catalytic arginine finger in the primary sequence, but the distal conformational effects in the α-helix may be predominant in the GAP structure, which is deduced from the crystallized structure (PDB ID: 3cxl). Several missense variants in the GAP domain have been identified in OPHN1, one of the X-linked intellectual disability genes (https://www.hgmd.cf.ac.uk/ac/index.php). G412D, M461V, and G529R mutations are causative for the intellectual disabilities, and mutated residues are located in the catalytic loop near the arginine finger, C-terminal $\alpha_5$, and the $\alpha_9$ helices, respectively [61–63]. In these cases, the mutational effect on GAP activities remains obscure. The Gly$^{847}$ residue in the $\alpha_9$ helix of PARG1 GAP domain that corresponds to the mutated site partially showed WT-specific motion distinct from mutant complexes (data not shown). The Gly$^{710}$ and Ile$^{755}$ residues of PARG1, corresponding to the mutated residues in OPHN1, showed similar motions in wild-type and mutated complexes in Ramachandran plot analysis, however, the substitution of the conserved glycine to charged residues might cause regional conformational changes in the GAP domain (S7 Fig).

Regarding the biological function of the C1 motifs, several structural studies have indicated that atypical C1 has synergistically diverse roles with an adjacent domain in the interaction with effector GTPases or structural integrity in the signaling pathway. For example, the C1 and PH domains of Vav1 RacGEF form contacts with the DH region as a structural unit, and restriction of its conformational flexibility leads to enhanced GEF activity [64]. The interaction of the C1 domain of the type V adenylyl cyclase with the small Gαi protein inhibits its enzymatic activity with inactive conformational catalytic changes for Gαs bound to the C2 region [65,66]. Interestingly, the predicted C1 domain of PARG1 that comprises of 4 β-sheets with α-helix formed interfaces to the C-terminal $\alpha_3$ helix of RhoA protein, with structural coordination of the conserved GAP interface residues. The Arg$^{616}$ and Thr$^{622}$ mutation to His$^{616}$ and Met$^{622}$ detected in NSCL/P are adjacent

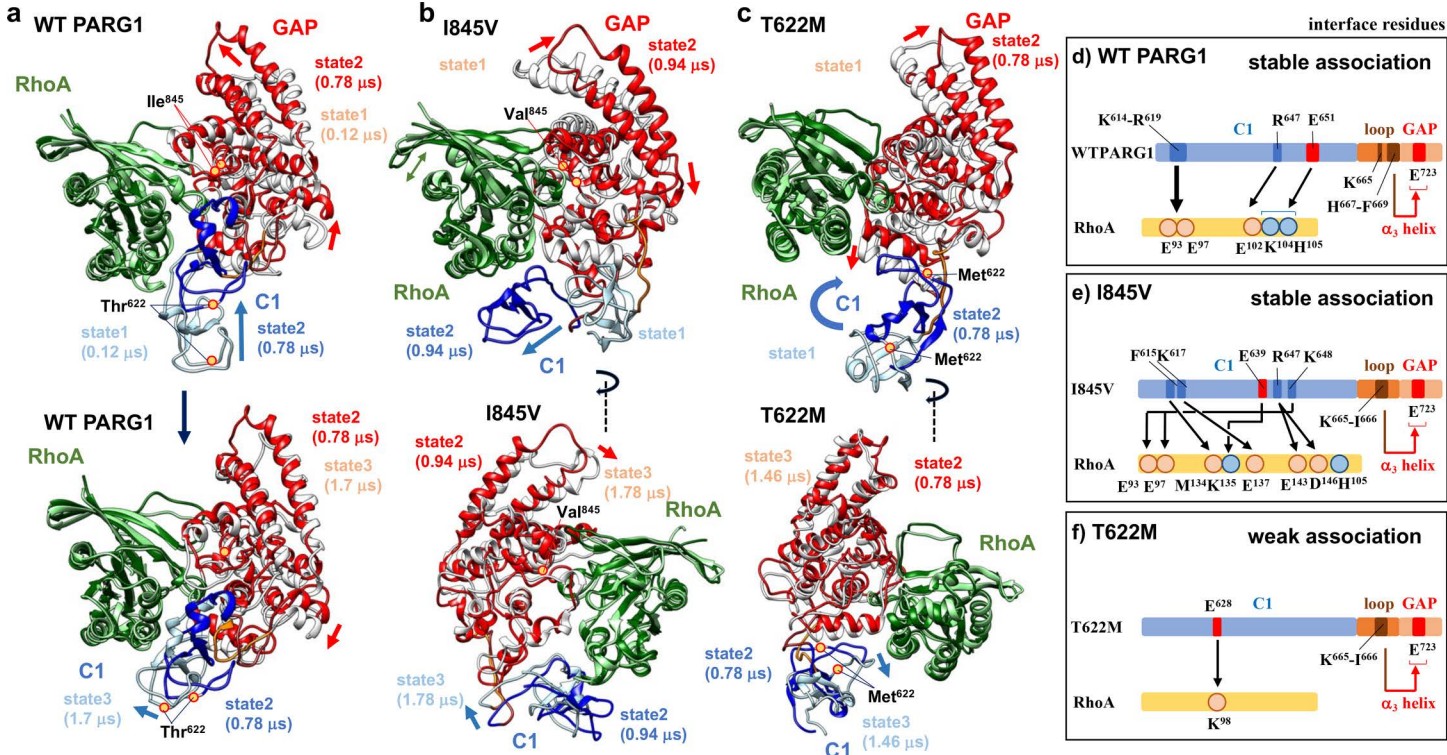

**Fig 9. Summary of structural dynamics of WT and mutant RhoGAPs-RhoA complexes.** (**a-c**) Representative structures of WT and mutant PARG1 RhoGAP complexes obtained by MD simulations were superimposed. (**a**) The C1-loop region and GAP domain of each PARG1 structure are indicated by the following color ribbons: WT PARG1 in state 1, at 0.12 μs in cluster 1 (0-0.12 μs) (upper panel) (C1: light blue; loop: orange; GAP: white), WT PARG1 in state 2, at 0.78 μs in cluster 5 (0.74-1.18 μs) (C1: blue; loop: brown; GAP: red), WT PARG1 in state 3, at 1.7 μs in cluster 7 (1.52-1.86 μs) (lower panel) (C1: light blue; loop: orange; GAP: white). Motions of state1−3 were selected from the representative structure with minimal RMSD in each cluster showing significant conformational changes (see Fig 4d–4f). (**b**) The C1-loop region and GAP domain of PARG1 I845V mutant complex: I845V in state1 (initial structure, C1: light blue; loop: orange; GAP: white), I845V in state 2, at 0.94 μs in cluster 1 (0.76-1.66 μs) (C1: blue; loop: brown; GAP: red), I845V in state3, at 1.78 μs in cluster 2 (1.74−2 μs) (initial structure, C1: light blue; loop: orange; GAP: white). (**c**) The C1-loop region and GAP domain of PARG1 T622M mutant complex: T622M in state 1 (initial structure, C1: light blue; loop: orange; GAP: white), T622M in state 2, at 0.78 μs in cluster 2 (0.6-1.0 μs) (C1: blue; loop: brown; GAP: red), T622M in state 3, at 1.46 μs in cluster 2 (1.38-1.74 μs) (C1: light blue; loop: orange; GAP: white). The positions of mutated residues are mapped as red circles on the superimposed structures. (**d-f**) Schematic mapping of the interface residues for RhoA substrate (yellow) in the C1 domain (blue) of PARG1 complexes. Interface residues between C1 region and RhoA protein of the PARG complexes are marked: the stable residue pairs forming hydrogen-bond/salt-bridges more than 100 ns period (5% of 2-μs simulation) are shown. (**d**) In WT PARG1 complex, C-terminal loop (His[667]-Phe[669] residues as shown in brown) interacted with the Glu[723] residue in the α3 helix (red) in the GAP domain. (**e, f**) In contrast, the corresponding residue in theα3 helix did not form stable hydrogen bonds with the C-terminal residues of the loop region in the mutant complexes.

to these predicted RhoA interface residues, and may also affect the regional interface structure of anti-paralleled β1-β4 backbone and coiled-coil structure in the C-terminal C1 domain.

The residue interface of the C1 domain of PARG1 for RhoA may play a significant role in its coordinated interaction with the substrate (Fig 9). Motions of the C1 domain of the wild-type PARG1 to interact with RhoA protein or the GAP domain were detected within initial phase during 2-μs MD simulation. MD simulations revealed that hydrogen bonding of the Asp[65] residue in the switch II of RhoA with a positively charged interface of α4 helix in the WT C1-GAP catalytic fold was conserved in the crystallized RhoA-bound p50RhoGAP complex (Fig 6b) [67]. In addition, hydrogen bonds between the RhoA Tyr[34] residue and Asn[841] residue in the α9 helix of WT GAP domain were conserved with the corresponding Asn[194] residue of the p50-RhoA complex, which stabilizes the loop region (Fig 6b–6d) [67]. The residues in C-terminal α9 helix

including the Ile[845] residue in the GAP domain formed stable hydrogen bond interactions each other, but not in the T622M mutant (Fig 6d), suggesting its effects on stable RhoA interaction. MD simulations also revealed that the altered angle torsion caused by the Ile[845] mutation to Val[845] may affect the $\alpha_3$-$\alpha_4$ helices and $\alpha_8$-$\alpha_9$ loop orientations in the catalytic fold of GAP domain, resulting in the distortion of C-terminal C1 conformation (Lys[648]-His[849] residues) of the C1-GAP domain (Figs 5d, 6a, S6a and S6e). MM-PBSA analysis suggested that alanine conversion of Ile[845] residue in WT PARG1 representative structure resulted in a significant reduction of the binding free energy for RhoA ($0.64 \pm 0.93$ kcal/mol). Therefore, its positional effect on the binding affinity to RhoA may be critical, in addition to its mutated effects on GAP structure. In 2-μs MD simulation, loss of coordinated C1 motion of I845V mutant, as seen in its N-terminal association with Met[134]-Glu[137] and Glu[143] residues in RhoA, maintained the complex, but the RhoA interface of the GAP region was disorganized especially in catalytic loop region including Arg finger (summarized in Fig 9). In this motion pattern, positively-charged residues of the GAP region with RhoA switch II region were detected similar to WT PARG1 complex (Fig 6b and 6c). The T622M mutant also showed unstable bond interactions between $\alpha_9$ helix and RhoA substrate (Fig 6c), and this may be caused by weak interaction of the C1 domain with RhoA. Unstable structural dynamics of the $\alpha_2$-helix (residue number: 686–698, Fig 7a-iii) and orientation of the $\alpha_3$ helix in the GAP domain may also induce distortion of RhoA interaction in T622M mutant (Fig 8). In addition, the deletion of loop region (residue number: 651–659) connecting the C1 and GAP domains in the WT complex caused GAP dissociation in the complex by short MD simulation of the representative structure at 0.78 μs, suggesting that the entire C1-loop region is important for the stable association of RhoA with the GAP domain (unpublished result). Therefore, C-terminal loop region has an important role for coordinated motion with the GAP domain in the complex (Fig 9d). As for expression analysis of the corresponding region of the C1-GAP domain (residue number: 611–886), the domain was not exogenously expressed in Neuro2a cells (unpublished result). This result was consistent with the report that YFP-tagged PARG1 was poorly expressed in HEK293 cells [68]. It is plausible that the presence of the N-terminal F-BAR domain may interfere with the motion of the C1-loop region, but dynamics of the domain organization was not in the scope of this analysis.

Recent structural analysis indicated that GTP-bound RhoA has an inactive conformation, and that switch II elicits a relatively low level of flexibility compared to the switch I region, which is distinct from the constitutively active form with reduced dynamics in the both switch I and II regions [69]. The unique dynamics equilibrium of switch I region in RhoA is considered to be important for the interaction of RhoGAPs. The RMSD values of the representative RhoA structure of WT PARG1 complex at 0.78 μs in MD simulation (Fig 5a) to the crystallized RhoA[G14V] structure in active conformation (PDB: 1a2b, [69,70]) was 3.35Å. In contrast, inactive conformation state of RhoA-GMPPNP (PDB: 6v6m) to the selected RhoA structure bound to the WT C1-GAP domain (Fig 1f) and the corresponding RhoA structure in the WT PARG1 complex were 1.41 Å and 3.04 Å of RMSD, respectively. The RMSD of inactive form of GDP-bound RhoA (PDB: 1dpf [71]) to the corresponding modeled RhoA structures showed 3.52 Å and 4.51 Å. MD analysis in this study showed that the wild-type C1-GAP interfaces may allow flexible motions of the switch I region in RhoA, which may adopt an inactive alternative open conformation of GTP-bound form of RhoA for hydrolysis [69]. These computational models of the C1-GAP domain are in line with the experimental studies and demonstrate how small topological changes caused by missense mutations affect the catalytic folding structure. Future experimental testing of mutational effects on GAP activity and PARG1 function in developing craniofacial tissues will help clinical interpretation of genetic variations in the context of craniofacial malformations.

## Supporting information

**S1 Fig. *in silico* alanine scanning of PARG1-RhoGTPase interactions.** Interacting residues in RhoGAP domains (a, c) of human WT PARG1[#] (residue number: 658−898), WT PARG1[##] containing the C1 domain (residue number: 611−886), I845V mutant, and T622M mutant interacting with RhoA (b, d) (PDB code: 5irc for PARG1[#]; 5c2k for PARG1[##], I845V, and T622M mutants) were examined by DrugScore[PPI] software program. Each of the interacting residues individually mutated

to alanine was scored by the change in binding free energy (ΔΔG) (a, b) and the degree of buriedness of each residue (c, d) in the interface was calculated. The residue number of target proteins is shown according to the modeled domain structure (Q52LW3−1).
(PDF)

**S2 Fig. Alignment of GAP domain sequences of PARG1 and p50RhoGAP, ArhGAP20 or p190ARhoGAP.** The alignments of GAP region of PARG1 RhoGAP with p50RhoGAP (PDB ID: 1ow3), ArhGAP20 (PDB ID: 3msx), and p190ARhoGAP (PDB ID: 5irc) that form complex with RhoA substrate. Multiple alignments were performed using MAFFT software and color code is based on Clustalσ scheme. The binding free energies were calculated using Drugscore$^{PPI}$ software and the interface residues for RhoA are indicated by circles. The signatures indicate the catalytic arginine finger (red arrowhead), the conserved interface residues (< 4Å) indicated by Amin et al. (2016) matched between PARG1 and each RhoGAP (filled black circle), the interface residues matched between PARG1 and each RhoGAP (black circle), and the conserved interface residues that showed different ϕ and φ angles in 2 µs MD simulation between wild-type PARG1 and missense PARG1 mutants (filled red circle). The Ile$^{845}$ residue locates as the predicted interface residue for RhoA in PARG1 that is matched to the corresponding interface residue of crystallized RhoGAPs.
(PDF)

**S3 Fig. Predicted structure model of C1-GAP domain of WT PARG1.** (a) RMSD of modeled C1-RhoGAP domain of WT PARG1 (residue number: 611−886) in 10-ns MD simulation. (b) RMSF of C1-RhoGAP domain (residue number: 611−886). 10-ns MD simulation was conducted and Phe$^{669}$ and Gly$^{670}$ played critical roles on the inter-residue interactions with the α$_3$-helix containing the conserved positively charged interface. The conserved interfaces for RhoA interaction are indicated as the follows: purple: C1 domain; orange: loop region; red: catalytic arginine finger; blue: positively charged α-helix; green: C-terminal interface. The conserved residues of PARG1 are mapped on the structural model (inset: brown: arginine finger; blue: positively charged interface; green: C-terminal interface). The arrowheads indicate the mutated sites identified in NSCL/P individuals (R616H, T622M, and I845V). (c) Position of mutated (red) and proximal (orange) residues that locate less than 5Å within the mutated residue (R616H, T622M, I845V) in NSCL/P in PARG1. Flexibility based on RMSF values is indicated (dark green: < 0.2; green: 0.2–0.6; pink > 0.6 in RMSF obtained from 10-ns MD simulation).
(PDF)

**S4 Fig. Interaction interface for modeled PARG1-RhoA complex structure.** The modeled RhoGAP domain with N-terminal C1 region was subjected to docking analysis using HDOCK in template-free mode. left panel: The interface residues (< 6Å) of WT PARG1 for RhoA binding were shown as overlay of the mapped interaction interface (gray) indicated in Amin *et al.* (2016) [2]. Conserved, variable residues of the GAP domain, and variable residues (variable1) near the arginine finger (variable2) are indicated as black, red, and blue, respectively. Lys$^{744}$ and Arg$^{748}$ residues are indicated as dark blue. The residue-residue contacts in the modeled PARG1 complex are shown in dark or light green color that matched the residues corresponding positions of the eight RhoGAP interface (see Fig 6B in [2]) or not as shown in Amin et al. (2016). right panel: The predicted interface residues of PARG1 for RhoA. The residue color of human PARG1 RhoGAP structure indicates as followed: purple: conserved region, blue: variable region 1, red: variable region 2. dotted circle: positively-charged interface (Lys$^{744}$ and Arg$^{748}$). The residues are mapped on the modeled structure according to Amin *et al.* (2016) [2].
(PDF)

**S5 Fig. Interaction interface for PARG1-RhoA complex structures modeled using AlphaFold-multimer.** The RhoGAP domains with N-terminal C1 region bound to RhoA were modeled using AlphFold-multimer program. The interface residues (< 6Å) of WT PARG1 for RhoA interaction were shown as overlay of the mapped interaction interface (gray) indicated in Amin *et al.* (2016) [2]. Complex model 1 and 3 were selected based on RMSD values to the crystallized RhoA

complex structures. The residue-residue contacts in the AlphaFold-modeled PARG1 complex are shown in dark or light green color that matched the residues corresponding positions of the eight RhoGAP interface (gray, see Fig 6B in [2]) or not as shown in Amin et al. (2016).
(PDF)

**S6 Fig. Ramachandran plot of the C1 loop region of WT and mutant PARG1.** The displayed angles were obtained from MD simulations of C1-RhoGAP/RhoA complexes in 2-μs MD simulation with Amber99SB-ILDN. Cys[657] (a), Leu[668] (b), and Phe[669] (c) residues were selected for WT PARG1-specific motions differed from I845V or T622 mutant complexed with RhoA. His[659] (d) and Lys[648] (e) residues in the C-terminal C1 domain showed diverse motions among RhoA complexes. Mutated Met[622] residue (f) showed angle motions partially distinct from Thr[622] residue of wild-type and I845V mutant bound to RhoA. The Lys[623] residue (g), the adjacent residues of Met[622], also showed T622M mutant-specific motions. The flexibility of an apposed β-sheet was affected by the point mutation, and its N-terminal loop region including Glu[640] (h) and Leu[642] (i) residues showed different motions from WT PARG1 complex. The Lys[648] residue (e) in I845V mutant showed specific motions distinct from WT-RhoA complex, since the residue interacted with RhoA in the mutant complex in 2-μs MD. These data were obtained from the following PARG1 C1-GAP ensembles. WT PARG1, left panel, I845V, Middle panel, T622M, right panel.
(PDF)

**S7 Fig. Hydrogen-bond profiles of interface residues in modeled PARG1-RhoA complex structures in 2-μs MD simulations.** The distribution of interface residues of modeled RhoGAP domains with N-terminal C1 region obtained by 2 μs-MD simulations are mapped: The hydrogen-bonds of interface residues in WT PARG1 or mutant PARG1 proteins and RhoA were shown in dark or light green color that matched the residues corresponding positions of the eight RhoGAP interface (see Fig 6B in [2]) or not as shown in Amin et al. (2016), based on overlay of the mapped interaction interface (gray) indicated in Amin *et al.* (2016) [2]. Conserved, variable residues of the GAP domain, and variable residues (variable1) near the arginine finger (variable2) in the RhoGAP domain are indicated as black, red, and blue, respectively. The pairwise residue matched or not matched to the corresponding positions of contact residues of the crystallized RhoGAPs (Fig 6B in [2]) are selected, based on the periods forming a residue pair more than 2 ns (0.1%: dark green) and 20 ns (1%: light green), respectively.
(PDF)

**S8 Fig. Ramachandran plot of the conserved residues in the GAP domain of WT and mutant PARG1.** The displayed angles were obtained from MD simulations of WT or mutant PARG1 C1-RhoGAP domains in 2-μs MD trajectories with Amber99SB-ILDN. It is noticeable that mutated residue Val[845] (a) in the complex showed motions distinct from Ile[845] residue in WTPARG1 or T622M mutant bound to RhoA. The ϕ and φ angles of the adjacent conserved Arn[841] (b) and Val[844] (c) residues and Ile[862] (d) and Leu[865] (e) residues in the $\alpha_{10}$-helix tended to show I845V-specific distortion in RhoA complex in 2-μs MD simulation. In T622M mutant, Glu[867] residue in the $\alpha_{10}$-helix showed distinct motions compared to other RhoA complexes. Arg[748] (g) and Gln[749] (h) in $\alpha_4$-helix residues were selected for WT PARG1 complex-specific motion angles differed from that of I845V or T622 mutant bound to RhoA. WT PARG1, left panel, I845V, middle panel, T622M, right panel.
(PDF)

**S9 Fig. Ramachandran plot analysis of the residues in the GAP domain showing WT-specific motions.** The displayed angles were obtained from 2 μs-MD simulations of WT or mutant PARG1 C1-RhoGAP domains. It is noticeable that Arg[852] (a), Thr[857] (b), Ile[860] (c), Arg[861] (d), Tyr[868] (e), and Arg[873] (f) residues in the $\alpha_{10}$-helix and its N-terminal loop region of WT PARG1 complex showed specific motions distinct from PARG1 mutants bound to RhoA. WT PARG1, left panel, I845V, middle panel, T622M, right panel.
(PDF)

**S10 Fig. Eigenvalue spectrum of RhoA protein bound to the C1-GAP domains of WT PARG1 and missense mutants.** (i) WT PARG1, (ii) I845V, (iii) T622M. The eigenvectors and eigenvalues were calculated by diagonalizing the atomic displacement correlation matrix of Cα atom coordinates from RhoA structures bound to each C1-GAP domain. (a) Screen plot shows the proportion of variance against the eigenvector rank. (b) Projection of trajectory to the planes of the first-two principal components.
(PDF)

**S11 Fig. Hydrogen-bond profiles of interface residues for PARG1-RhoA complex structures using HDOCK and AlphaFold-multimer programs.** The mapping of interface residues of modeled RhoGAP domains with N-terminal C1 region obtained by each MD simulation: The hydrogen-bonds of interface residues in RhoA-WT PARG1 using HDOCK in 2-μs MD (a) and modeled PARG1 obtained by AlphaFold multimer (b, c) were shown as overlay of the mapped interaction interface (gray) indicated in Amin *et al.* (2016) [2]. As for AlphaFold-modeled complexes, the profiles of model 1 and 3 were calculated from the trajectories of the stable complexes in 70 ns and 75 ns trajectories in MD simulations, respectively. The residue-residue contacts in the modeled PARG1 complex are shown in dark or light green color that matched to the residues corresponding positions of the eight RhoGAP interface (gray, see Fig 6B in [2]) or not as shown in Amin et al. (2016).
(PDF)

**S1 Table. HHPred template search summary.** Details of hits for the selected template of human PARG1 GAP domains are indicated. PARG1 (residue number: 671–886): the GAP domain retrieved from InterPro; PARG1#: GAP domain containing the N- and C-terminal loop (residue number: 658–898); PARG1##: GAP domain containing the C1 domain (residue number: 611–886). Prob: Probability of template to be a true positive. E(expect)-value: average hit of false positives. Score: raw score calculated by comparing the amino acid distributions between columns from the query alignment and columns from the template alignment. The probabilities for insertions and deletion at each position in the alignment are taken into accounts as positional specific gap penalties. SS: secondary structure score. Cols: The number of aligned match-match column in the HMM-HMM alignment. Query HMM: Range of query match states aligned. Template HMM: range of template match states aligned and, in parenthesis, total number of template HMM. Id(%): sequence identity. 3cxl.pdb: Structure of chimerin1. 5c2k.pdb: Structure of MgcRacGAP bound to RhoA. 2mbg.pdb: Structure of RalA-binding protein 1.
(PDF)

**S2 Table. Structural evaluation of RhoGAP domains.** The selected RhoGAP domains of human PARG1 were evaluated using PROCHECK, ERRAT and ProSA programs. Predicted GAP domain structures retrieved from InterPro search are shown with the template PDB structures in parenthesis. RhoGAP domains containing the N- and C-terminal loop regions (#, residue number: 658–898, 241 amino acids) and the C1 domain (##, residue number: 611–886, 276 amino acids), defined using InterPro software program, were also assessed by the same procedures.
(PDF)

## Acknowledgments

The authors thank helpful discussion on the simulation environment construction on "Flow" Type II subsystem. The MD simulation of AlphaFold-modeled PARG1 complex was carried out using supercomputer "Flow" Information Technology Center, Nagoya University.

## Author contributions

**Conceptualization:** Zen Kouchi.

**Investigation:** Zen Kouchi, Masaki Kojima.

**Writing – review & editing:** Zen Kouchi.

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
