## [Decision Letter · Decision Letter 0]

Dear Dr. Kouchi,

We look forward to receiving your revised manuscript.

Kind regards,

Paul A. Randazzo

Academic Editor

PLOS ONE

Journal Requirements:

3. We note that your Data Availability Statement is currently as follows: [All relevant data are within the manuscript and its Supporting Information files.] Please confirm at this time whether or not your submission contains all raw data required to replicate the results of your study. Authors must share the “minimal data set” for their submission. PLOS defines the minimal data set to consist of the data required to replicate all study findings reported in the article, as well as related metadata and methods (https://journals.plos.org/plosone/s/data-availability#loc-minimal-data-set-definition). For example, authors should submit the following data: - The values behind the means, standard deviations and other measures reported; - The values used to build graphs; - The points extracted from images for analysis. Authors do not need to submit their entire data set if only a portion of the data was used in the reported study. If your submission does not contain these data, please either upload them as Supporting Information files or deposit them to a stable, public repository and provide us with the relevant URLs, DOIs, or accession numbers. For a list of recommended repositories, please see https://journals.plos.org/plosone/s/recommended-repositories. If there are ethical or legal restrictions on sharing a de-identified data set, please explain them in detail (e.g., data contain potentially sensitive information, data are owned by a third-party organization, etc.) and who has imposed them (e.g., an ethics committee). Please also provide contact information for a data access committee, ethics committee, or other institutional body to which data requests may be sent. If data are owned by a third party, please indicate how others may request data access.

4. We note that you have referenced (unpublished data) on page 16, which has currently not yet been accepted for publication. Please remove this from your References and amend this to state in the body of your manuscript: (ie “Bewick et al. [Unpublished]”) as detailed online in our guide for authors

Reviewers' comments:

Reviewer's Responses to Questions

**Comments to the Author**

1. Is the manuscript technically sound, and do the data support the conclusions?

Reviewer #1: Yes

Reviewer #2: Partly

2. Has the statistical analysis been performed appropriately and rigorously?

Reviewer #1: Yes

Reviewer #2: Yes

3. Have the authors made all data underlying the findings in their manuscript fully available?

Reviewer #1: Yes

Reviewer #2: Yes

4. Is the manuscript presented in an intelligible fashion and written in standard English?

Reviewer #1: Yes

Reviewer #2: Yes

Reviewer #1: Reviewer Report

Title:

Structural Analysis of the PARG1 RhoGAP Domain in Relation to NSCL/P Mutations: A Computational Perspective

Summary of the Study

The manuscript presents an extensive computational investigation into the structural dynamics and functional implications of PARG1 RhoGAP domain mutations (T622M and I845V) implicated in NSCL/P. The author employed advanced computational tools, including homology modeling, molecular docking, and molecular dynamics (MD) simulations, to characterize wild-type (WT) and mutant PARG1 domains. The results suggest that these mutations disrupt the RhoA-binding interface and alter protein stability, providing mechanistic insights into their potential pathogenicity. The use of diverse computational techniques ensures a thorough structural and functional evaluation of WT and mutant PARG1 RhoGAP domains. The manuscript provides an in-depth characterization of key residues, hydrogen bonding networks, and conformational dynamics. However, there are several issues required to be addressed by the author.

Major comments:

• The manuscript describes the use of traditional computational techniques, including homology modeling and molecular docking, to model the protein-protein interactions involving the disease-relevant mutations in the PARG1 RhoGAP domain. However, it is notable that the author did not utilize AlphaFold, a highly accurate and widely accepted tool for protein structure prediction, particularly for complex structures. The author could have considered using AlphaFold-Multimer to generate the protein-protein complex models, especially for validating the interactions between PARG1 and its partners. This would not only have strengthened their findings but also added confidence to their molecular dynamics simulations, as AlphaFold-generated structures tend to serve as better starting points for downstream simulations due to their higher accuracy.

• The manuscript reports performing MD simulations for 100 ns, which may be considered relatively short for accurately capturing the full scope of protein dynamics and complex interactions, particularly for systems involving protein-protein interactions or allosteric effects. In many cases, microsecond-scale simulations provide a more comprehensive view of the system's behavior, allowing for a deeper understanding of long-range conformational changes, transient interactions, and more stable equilibrium states. Protein-protein interactions and allosteric effects, for instance, may not fully manifest over just 100 ns, as significant changes in conformation and the establishment of stable interaction networks can take longer to develop. Rare events, such as the binding/unbinding of ligands or conformational transitions, might require longer simulation times to be adequately sampled.

• While the manuscript provides a solid computational analysis of the structural changes caused by the T622M and I845V mutations, it lacks a detailed discussion on how these findings correlate with published biological data. The connection between these computational observations and known experimental studies on the effects of similar mutations in RhoGAP proteins (or homologous proteins) is not adequately addressed. Mutations in the RhoGAP domain are known to affect the regulation of RhoA GTPases, a key player in cellular processes such as migration and cytoskeletal organization. Including references to experimental studies linking RhoGAP mutations to cellular dysfunction would strengthen the manuscript and provide biological context.

• The manuscript introduces an intriguing link between PARG1 RhoGAP mutations and NSCL/P, but the connection remains speculative. The discussion could benefit from more focus on the developmental biology of cleft lip and palate and how RhoA signaling is involved in craniofacial development. Altered cellular dynamics resulting from disrupted RhoA regulation are likely contributors to developmental disorders, but this connection is not sufficiently explored.

• MMPBSA analysis is a useful tool for estimating the binding free energy differences between mutant and wild-type protein-protein complexes. In the context of the T622M and I845V mutations, this method could help quantify how these mutations affect the interaction stability between PARG1 and RhoGAP, potentially linking changes in interaction energy to disease-causing effects.

• The RMSD plots presented in the manuscript lack clarity regarding their generation, raising questions about the apparent discontinuities. These breaks could result from improper post-processing of trajectory data, potentially due to periodic boundary conditions (PBC) not being correctly accounted for during the RMSD calculations. Proper re-centering and imaging should be performed to ensure a continuous and accurate trajectory representation. Furthermore, the figure legends are overly descriptive but fail to provide sufficient explanation for critical features in the plots. For instance, the annotations "1," "2," and "3" in the RMSD plots are introduced without a clear description of what they signify. The author should revise the legends to concisely explain these elements and ensure the figures are intuitive and informative.

Minor Comments

• Several sections in the manuscript contain repetitive information, especially regarding the interactions of specific residues with RhoA and their role in protein stability. This redundancy can detract from the clarity of the manuscript.

• The figures are well-designed and aid in understanding the computational results. However, some figure legends are repetitive and could be streamlined to highlight the most important findings.

• Y-axis label units (Angstroms??) are missing in Fig. 8a.

Reviewer #2: Overview

In this work, Dr. Kouchi tests the hypothesis that the C1 domain of PARG1 (ArhGAP29) regulates the protein’s GAP function on RhoA. Specifically, the work supports the idea that the C1 domain reversibly binds directly to RhoA, which stabilizes interactions between the GAP domain and RhoA. These stabilized interactions are proposed to facilitate the movement of RhoA switch I, which is necessary for GAP-mediated catalysis. Furthermore, the C1-GAP:RhoA interactions appear to be disrupted by disease-causing nonsynonymous mutations in the C1 domain (T622M) and in the GAP domain (I845V). The mechanisms by which the mutations cause GAP dysfunction—which has not previously been understood—are proposed and supported by in silico efforts.

Major Issues

1. The major shortcoming of the paper is that all of the data is simulated, not experimentally validated. For example, demonstrating that the C1-GAP construct of PARG1 (residues 611 – 886) exhibits greater catalytic activity than the GAP domain with N- and C-terminal ends extended (residues 658 - 898) would be appropriate, as the author argues that the C1 domain modulates GAP activity through interdomain (GAP) and substrate (RhoA) interactions. Furthermore, testing the activities of I845V and T622M C1-GAP constructs would corroborate the author’s proposal that their catalytic efficiencies are decreased compared to WT enzyme. In lieu of these experiments, the author should strongly reiterate in the abstract and discussion that their findings are solely based on molecular modeling, docking, and dynamics, and that these findings need to be experimentally validated in the future.

2. Although touched upon briefly, a thorough analysis of the R616H mutation was not performed. Could the author comment on why this was not done, or instead, add these analyses to the manuscript?

Minor Issues

General comments

1. "Hydrophilic" seems to sometimes be used instead of "hydrophobic," please check each usage on a case-by-case basis and ensure that the correct term is used. For example, many residues within “hydrophilic” interfaces appear to be those that are classically considered to be hydrophobic.

2. Figures have so many labels that it is difficult to distinguish individual features. Consider removing ones that are unnecessary to convey the point (e.g., many amino acid labels in figures 1, 2, 3, 6, and 6).

3. Statements saying "conserved" and "matched" are not always clear, please use in a more quantitative sense. What does "highly conserved" and "well-matched" mean? Similarly, regions of the GAP are referred to as "conserved variable" or "variable conserved" regions, which is not intuitive to the reader as these have opposing meanings.

4. Might the patterns of C1 domain interactions depicted in Fig. 5A (arising from MD simulations) be altered in the full-length protein? I.e., if the F-BAR and linker region to C1 domain were present, would it restrict the movement and cause a preference for either pattern? In other words, how biologically relevant are the two patterns that are proposed?

5. Might the mutational effects of T622M not stem from changes in the dynamics of the α4 and α9 helices as indicated on page 13, but instead simply from steric clashes caused by the methionine that prevent either pattern A or B (Fig. 5A) from forming? T622M seems to be localized near the interfaces in both patterns, particularly pattern B (although it is hard to tell from the figure).

6. It would be appropriate to add a cartoon summary figure showing the localization of the C1 domain relative to GAP domain and bound RhoA substrate in all MD simulations (i.e., patterns A and B in WT, and mobile in mutant PARG1). Furthermore, the summary figure should succinctly explain the major effects caused by each mutation (for I845V, destabilization of N841 contacts with concomitant movement of C1 domain leading to loss of interactions/affinity to RhoA switches I and II; and for T622M, due to altered interface of C1 domain to RhoA and subsequent movements of C1-linker and GAP helices α4 and α9, leading to loss of RhoA switch I contacts). This would make the overarching findings of the study much easier to follow.

Introduction

• Line 1: as written, Rho GEFs appear as though they regulate GTP hydrolysis rather than exchange. Please correct.

Results

• Page 10, R748 does not appear to be conserved in 5irc (p190ARhoGAP) but is stated to be a critical residue for the RhoA interface?

• Page 10, "The calculation using DrugScorePPI showed that predicted interface residues of PARG1 GAP domain containing the C1-domain were well-matched to those of the other RhoGAP complexes bound to RhoA, compared to the RhoA bound GAP domain with the N-and C-terminal regions." There doesn't appear to be much of a difference between PARG1# and PARG1## in Fig. S2?

• Page 10, this is the first mention of residue R616 other than in supplemental figures, please introduce this mutated residue sooner (e.g., in Introduction).

• Page 10, "point mutation at Thr622 affected a more flexible region than the Arg616 residue in the C1-GAP domain during the 10-ns MD simulation (Fig. 1d, Fig. S3b)." Perhaps this is meant to cite Fig. 1F rather than 1D?

• Page 11, how is the theoretical binding affinity of I845V calculated to be 10-fold lower than WT when the binding free energies are unchanged? As for T622M, Fig. 4 suggests that the periods in which RhoA is dissociated from the GAP are increased for both mutants, which would perhaps suggest a lower affinity; is the 100 ns simulation representative of apparent affinity?

• Page 12, "The flexibility of the His659 and Lys648 residues reflected the stability of the RhoA-bound complex, as a wide range of �-� angles were detected in the stable complex formed during the 100-ns simulation. (Fig. S5b, c, g)." Please clarify, how does the flexibility of these residues indicate stability of the GAP:RhoA complex?

• Page 13, "Mutational effects of T622M and I845V were attributed to the C-terminal positively-charged α4-helix interface (Fig. S6b and S6c) and the N- and C-terminal edges of the α9-helix of the GAP domain (Fig.6a, Fig. S6d-S6f)." Please highlight these regions in the main Fig. 6A and/or cite the relevant residues in the main text.

• Page 13, "In pattern B of WT PARG1, the Asn841 residue in the hydrophilic interface formed stable hydrogen bonds with the Ile845 residue and switch I region of RhoA, suggesting that theoretically conserved GAP interfaces properly recognize the RhoA protein through cooperative motions of the C1-linker domain (Fig. 6c, Fig. S4)." How are the hydrogen bonds of N841 with I845 and switch I of RhoA indicative of cooperative motions of C1-linker in the WT context?

• Page 14, typo: "The β1-sheet of the C2 domain" should say "C1 domain," correct? Similarly, later on the page, "The N-terminal motion of the C2 domain..." should say “C1 domain.”

• Page 14, same sentence: "The β1-sheet of [C1] domain...was closed to the C-terminal α3-helix..." Could the author clarify what "closed" means?

• Page 15, switch I of RhoA appears to be mobile in one RMSF trace in I845V mutant (Fig. 8A iii), but this isn't reflected in the principal component analysis (Fig. 8C and 8D)?

Discussion

• Page 16, could the author comment more on the ways in which the AlphaFold model-derived docking was different than reported literature?

• Page 18, last paragraph: the discussion centered on switch I and how its mobility is necessary for catalysis should be elaborated on (perhaps this could be a part of the summary figure, see General comment #6).

Figures

• S1: Please make labeling of PARG1* and WT PARG1 the same as elsewhere in manuscript (i.e., PARG1# and PARG1##). Also, please make the residue ranges shown for RhoA in panels B and D consistent, the ranges shown for WT PARG1 (or PARG1##) is truncated at 102 versus 134. Main text (page 9) indicates that RhoA from 5irc is used for this analysis, but legend states 5c2k; please clarify.

• S2: Please make labeling of PARG1* and WT PARG1 the same as elsewhere in manuscript (i.e., PARG1# and PARG1##). Please indicate where residue 845 is located, and add more numbers to each line, "700" is not sufficient for the reader. The circles (filled black, black, and red) are not very easy to interpret in terms of meaning, please re-write legend to explain more clearly (e.g., what is different between filled black and black circles?).

• S3B: Please label C1 and GAP domains.

• S4: The left panel is not very intuitive, for example what do the colors on the left-hand side (RhoA) represent? As for the squares, what is the difference between dark and light green? The legend indicates that they are "corresponding positions of the eight RhoGAP interface or not," but what eight interfaces are we referring to? On the right, the color coding is not accurate, conserved residues are shown as purple (not black as stated in legend). Does HsPARG1 refer to human PARG1? Please use consistent nomenclature.

• S5: These panels should be shown in the order they are introduced in the main text (e.g., the first ones discussed are panels D - F, which should therefore be A - C). For the current panel A, an extra label "(Met622)" is present on T622M charts, please remove. The figure legend for panel G is labeled as "vii", please fix.

• 1A: Consider adding locations of mutations discussed in the work.

• 1C, 1D: Should "hydrophilic" in figure legend be "hydrophobic"? Many residues are hydrophobic.

• 1E: Please color C1 domain differently than RhoA in Fig. 1D and 1G.

• 1F: X-axis legend should be amino acids of PARG1 (611 – 886), not construct. Red and brown are hard to differentiate in the figure.

• 2: Please label some of the regions of the GAP domain, it is difficult to compare to the predicted binding interface shown in Figure 1G. Which helices are being shown? Further, what are the black residues; are they necessary to show?

• 3C: Please show and label residue I845.

• 4: The figure legend is difficult to parse and some of the information seems like it would be more appropriate to put into the main text. Please also add units to the X-axes (presumably, nanoseconds). There may also be a typo, does the 0.772 Å magnitude refer to a-1 instead of b-3?

• 5A: What does red and blue color coding in upper left panel (initial complex structure) represent? Similar to Fig. 4, some of the information in the legend may be more pertinent to the results section. There is also a typo in legend, Arg 625 is listed as "Lys.”

• 5B: What does the color coding of the boxes represent? If charge, should histidine be blue?

• 6B, 6C: Similar to fig. 5B, please state what the color codes of the boxes represent (e.g., P36 is orange, but why?). Also, it would be much easier to understand the differences between networks if the residues lined up (e.g., the first residue shown for RhoA in each row is V33, D13, G14, and A15, which is confusing). As it is, one struggles to discern the major differences in interaction networks. Perhaps only the major changes in networks should be shown as another panel (i.e., I845V versus pattern B and T622M versus pattern B)?

• 7A: Please use residue numbers of PARG1 (611 – 886), not construct.

• 7B iv: The RMSF scores for T622M look to be higher in C1 domain (orange residues) than shown in 7A iv (RMSF < 1). Could the author comment on this discrepancy?

• 8: Please indicate in titles for WT PARG1 which pattern (A or B from Fig. 5) is shown (i.e., for 8A i and ii), similar to 7A i and ii.

Tables

• S1: Please add residue ranges for PARG1 in legend to match PARG1# and PARG1##.

**Do you want your identity to be public for this peer review?** For information about this choice, including consent withdrawal, please see our Privacy Policy

Reviewer #1: No

Reviewer #2: No

---

## [Author Response · Author response to Decision Letter 1]

3 Mar 2025

Dr. Paul A. Randazzo

Academic Editor

PLoS ONE

Revision of manuscript, submission ID PONE-D-24-50037 [EMID:1ae5aec5f3027c64]

Dear Dr. Randazzo,

Please find attached revised manuscript and marked-up copy of the manuscript that highlights changes made to the original version by Kouchi and Kojima entitled “C1-linker region of PARG1 RhoGAP promotes the catalytic recognition fold of RhoA substrate“, as research article “PLoS ONE”. We read carefully your recommendations and the reviewer’s criticism and revised the paper accordingly. Here is our point-to-point response to the reviewers’ comments and a discussion of all changes made in the revised manuscript.

Editor’s comment

(1) The duration of the simulations (100 ns) is important to address. Simulations of at least 2-micro sec should be used and these might not be adequate for examining protein complexes that can rearrange on a much longer time scale. The importance of time scale should be discussed in addition to extending the simulations for as much time as reasonably possible.

(2) Using AlphaFold to determine starting structures for MD simulations of the PARG1.

(1) We obtained similar conformational change of the wild-type PARG1-RhoA complex as shown in pattern B (Figure 5) in the original article by conducting another MD-run. We also performed 2-micro s scale of MD simulation of the wild-type and I845V PARG1 complexes, and the results were added to the revised manuscript in Figure 4-9 and Supplementary Figures 5-7 and 9. 2. The importance of time scale in the complex was discussed in the 5th paragraph of Discussion part of the revised manuscript. (2) The modeling of AlphaFold was performed but the docking analysis using HDOCK was not consistent with the experimental result that showed specific interaction with RhoA (see Ref. 3 and 10), MD simulations of the selected RhoA-bound model based on AlphaFold multimer was performed, but stable complex was not obtained more than 50-75 ns through long-ranged simulation analysis: irreversible dissociation patterns of the complex were detected. The details on the complex were added to Fig. 1h and in the result and discussion parts of the revised manuscript, but the result of MD simulation was not included in the manuscript.

To Reviewer1

To Major Comments

1-1: The manuscript describes the use of traditional computational techniques to model the protein-protein interactions involving the disease-relevant mutations in the PARG1 RhoGAP domain, however, it is notable the author did not utilize AlphaFold, a highly accurate and widely accepted tool for protein structure prediction, particularly for complex structures. The author could have considered using AlphaFold-Multimer to generate the protein-protein complex models, especially for validating the interactions between PARG1 and its partners.

(Answer) The corresponding region of PARG1 was structurally modeled using AlphaFold software. Cdc42(PDB code: 5c2j) was candidate as substrates for both the AlphaFold-modeled RhoGAP retrieved from InterPro and C1-RhoGAP domains using the HDOCK template-free mode. However, the RhoA-bound forms was previously reported in mono-specificity for RhoA of PARG1 in several papers (Ref. 3, 10). Therefore, RhoA-bound form was also selected using AlphaFold multimer (ColabFold) (Ref. 29,30) and the selected model showing minimum value of RMSD with analogy to the crystallized RhoGAP-RhoA complex (PDB ID: 5c2k) was further analyzed for MD simulation. RhoA structure was predicted based on 2-185 amino acids, corresponding to the used coding region of RhoA as the structure, i.e. PDB ID: 5c2k, for docking.

1-2: This would not only have strengthened their findings but also added confidence to their molecular dynamics simulations, as AlphaFold-generated structures tend to serve as better starting points for downstream simulations due to their higher accuracy.

(Answer) Structure of modeled C1-RhoGAP bound to RhoA using AlphaFold multimer (ColabFold) was selected by comparing MgcRacGAP complex structure (ePDB ID: 5c2k). Several analyses on the selected AlphaFold-based structural model did not show any stable motions (50-75ns periods) as the RhoA-bound complex for long simulation period and low levels of dynamics of the C1 domain were detected. Therefore, the long-ranged conformational analysis of the modeled complex was not conducted. Details in the model was mainly described in the Fig. 1h, Result section and the 2nd paragraph of the Discussion part.

2-1: The manuscript reports performing MD simulations for 100ns, which may be considered relatively short for accurately capturing the full scope of protein dynamics and complex interactions, particularly for systems involving protein-protein interactions or allosteric effects. In many cases, microsecond-scale simulations provide a more comprehensive view of the system’s behavior, allowing for a deeper understanding of long-range conformational changes, transient interactions and more stable equilibrium states.

(Answer) Motion property of the C1-GAP domain complex of wild-type PARG1 and its mutant effects in the GAP region on the interaction dynamics of RhoA proteins were especially important issues, therefore the modeled complexes were analyzed for 2-�s MD simulation, using Desmond Molecular Dynamics system. The long-range conformational changes of wild-type PARG1 complex were similar to the dynamics detected in 100ns-MD simulation, as categorized by “pattern B” in the Figure 5. N-terminal positively-charged residues (Lys614, Arg616, Lys617, and Arg619 residues) in the C1 domain formed hydrogen bond interactions with Glu93 and Glu97 residues in the RhoA substrate. In contrast, the corresponding N-terminal C1 domain (Phe615 and Lys617 residues) in I845V formed stable hydrogen bond interactions with more C-terminal region (Met134 and Glu137 residues) of RhoA protein. Glu639 and Arg647-Lys648 residues in the C1 domain also mediated the interaction with Lys135 and Glu143/Glu93 residues, respectively. Principal component analysis revealed that this C1 domain-mediated interaction caused wide range of flexibility in the N-terminal portion of RhoA, in contrast to the motion of wild-type PARG1 complex as shown in Fig. 8h. The catalytic Arg707 residue of I845V mutant complex was very close to the several residues located in switch I and II regions (Pro31, Glu32, Val35, Thr37, Asp59, Gln63 residues) more than 20ns periods in transient state. In addition, Lys714 residue in the C-terminal catalytic loop of the mutant complex formed hydrogen bond with Asp100 residue of RhoA (see Figure S4). The summary of the hydrogen interaction network is indicated for wild-type PARG1 complex in Fig. 6b, c in the revised figure, and the motion dynamics of I845V mutant complex was described in the Discussion part.

2-2: Protein-protein interactions and allosteric effects, for instance, may not fully manifest over just 100ns, as significant networks can take longer to develop. Rare events, such as the binding/unbinding if ligands or conformational transitions, might require longer simulation times to be adequately sampled.

(Answer) The conformation of the RhoA complex indicated in Fig. 5a-B was not one of the rare events in that the C1 domain formed the interface to RhoA substrate. Another trajectory with similar structural motion of the complex with Fig. 5a-B has also been obtained using the same condition of 100-ns MD simulation.

3-1: While the manuscript provides a solid computational analysis of the structural changes caused by T622M and I845V mutations, it lacks a detailed discussion on how these findings correlate with published biological data. The connection between these computational observations and known experimental studies on the effects of similar mutations in RhoGAP proteins (or homologous proteins) is not adequately addressed.

(Answer) Phenotype of I845V mutation was reported by the following papers, Leslie et al. (2012) Birth Defects Research. Part A, Clinical and Molecular Teratology and Tang et al. (2020) BioMed Res. International. As shown in the paper, details in physiological or biochemical effects of the mutation were not described. YFP-tagged PARG1 was reported to be poorly expressed in HEK293 cells (Anderson EL and Hamann MJ, Biochem. J. 441: 869-871, 2012) and difficulties in cellular studies of PARG1 have been suggested. All the references on the variants are included in the revised version.

3-2: Mutations in the RhoGAP domain are known to affect the regulation of RhoA GTPases, a key player in cellular processes such as migration and cytoskeletal organization. Including references to experimental studies linking RhoGAP mutations to cellular dysfunction would strengthen the manuscript and provide biological context.

(Answer) The reported experimental studies linking RhoGAP mutations were added to the 3rd paragraph of Discussion part with additional references. All the listed references are referred to HGMD database (http://www.hgmd.org), and pathological missense mutations within the GAP region associated with cytoskeletal disorganization have been reported in the following RhoGAP proteins: ArhGAP24 (R142C, Q158R), Oligophrenin1(OPHN1) (G412D, M461V, Q529R), N-chimerin1(CHN1) (E313K).

4: The manuscript introduces an intriguing link between PARG1 RhoGAP mutations and NSCL/P, but the connection remains speculative. The discussion could benefit from more focus on the developmental biology of cleft lip and palate and how RhoA signaling is involved in craniofacial development. Altered cellular dynamics resulting from disrupted RhoA regulation are likely contributors to developmental disorders, but this connection is not sufficiently explored.

(Answer) Physiological roles of RhoGTPase signaling including RhoA, associated with cytoskeletal reorganization in craniofacial development, were added to the 1st paragraph of Discussion part with references. Pathological RhoA activation in cleft cells from NSCL/P patients are known to cause reduced cell motility with Cdc42 activation, but coordinated activation of several RhoGTPases in neural crest cells and craniofacial mesenchyme during embryonic development has been experimentally validated using conditional knockout or dominant negative Rho kinase mutant models.

5: MMPBSA analysis is a useful tool for estimating the binding free energy differences between mutant and wild-type protein-protein complexes. In the context of the T622M and I845V mutations, this method could help quantify how these mutations affect the interaction stability between PARG1 and RhoGAP, potentially linking changes in interaction energy to disease-causing effects.

(Answer) MMPBSA analysis was performed based on the representative trajectories considering 2D-RMSD analysis and these data were added to the manuscript as Table 1. Wild-type PARG1-RhoA complex in pattern B (Fig. 5) showed lower binding free energy (-56.02 kcal/mol in No.1; -79.64 kcal/mol in No.2 in Fig. 4) than those of T622M (-33.40 kcal/mol in No.1; -48.47 kcal/mol in No.2) and I845V complexes (-48.54 kcal/mol in No.1; -38.11 kcal/mol in No.2), indicating that the C1 domain promoted the stable interaction of the GAP domain with RhoA protein. The mode of stable interaction or irreversible interaction during 100ns MD simulations did not reflect the values of binding free energies of mutant PARG1-RhoA complexes obtained by MMPBSA, suggesting that the conformation of RhoGAP domain in the complex, in addition to its affinity to the RhoA substrate, may regulate their stability of RhoA interaction. This conclusion was also supported by the similar value of binding free energies of wild-type PARG1-RhoA complex (-31.97 kcal/mol) in pattern A (Fig. 5) to those of mutant PARG1 complexes.

6-1: The RMSD plots presented in the manuscript lack clarity regarding their generation, raising questions about the apparent discontinuities. These breaks could result from improper post-processing of trajectory data, potentially due to periodic boundary conditions (PBC) not being correctly accounted for during the RMSD calculations. Proper recentering and imaging should be performed to ensure a continuous and accurate trajectory representation.

(Answer) RMSD values of the RhoA-dissociated complex was corrected in Fig. 4 based on the conditions in the simulation box with 8 Å in minimum distance between the box and the protein. The data on long-ranged MD simulation analysis was also added to the Fig. 4.

6-2: Furthermore, the figure legends are overly descriptive but fail to provide sufficient explanation for critical features in the plots. For instance, the annotations “1”, “2”, and “3” in the RMSD plots are introduced without a clear description of what they signify. The author should revise the legends to concisely explain these elements and ensure the figures are intuitive and informative.

(Answer) The MD trajectory was selected based on summation of the duration (ns) of the stable complex and the trajectories showing minimum period of dissociated forms was presented as the representative trajectories.

To minor comments

Several sections in the manuscript contain repetitive information, especially regarding the interactions of specific residues with RhoA and their role in protein stability. This redundancy can detract from the clarity of the manuscript.

(Answer) The redundancy for the description on the interface residues and stability was avoided by changing the presentation of figures and description in the revised manuscript.

The figures are well-designed and aid in understanding of the computational results. However, some figure legends are repetitive and could be streamlined to highlight the most important findings.

(Answer) The main points for the analysis were concisely presented as the revised figure, and summarized schemes of the motion analysis was added as the Figure 9.

Y-axis label units are missing in Fig. 8a.

(Answer) Y-axis unit was labeled in Fig. 8a in the revised manuscript.

To Reviewer2

To Major Comments

1: The major shortcoming of the paper is that all of the data is simulated, not experimentally validated. For example. Demonstrating that the C1-GAP construct of PARG1 (residue 611-886) exhibits greater catalytic activity than the GAP domain with N-and C-terminal ends extended (residues 658-898) would be appropriate, as the author argues that the C1 domain modulates GAP activity through interdomain (GAP) and substrate (RhoA) interactions. Furthermore, testing the activities of I845V and T622M C1-GAP constructs would corroborate the author’s proposal that their catalytic efficiencies are decreased compared to WT enzyme. In lieu of these experiments, the author should strongly reiterate in the abstract and discussion that their findings are solely based on molecular modeling, docking, and dynamics, and that these findings need to be experimentally validated in the future.

(Answer) The aim performing molecular modeling and molecular dynamics analysis, but not experiments, was added to Abstracts and the 2nd and last paragraph of Discussion part.

2: Although touched upon briefly, a thorough analysis of the R616H mutation was not performed. Could the author comment on why this was not done, or instead, add these analyses to the manuscript?

(Answer) R616H mutation was reported in 55 individuals with NSCL/P cases from the Philippines as described in Table1 of Leslie’s paper (published in 2012, Ref 4), but 34 control samples include this variant as a rare variant category, which is in contrast to the T622M and I845V NSCL/P cases only, categorized as very rare variants. The Arg606 locates the 1st �-sheet in the predicted structure of the C1 domain (Fig. 3) and both of the Arg and His residues possess similar amphiphilic properties with the long side chains. Therefore, The R616H mutation was not selected for the analysis and structural role of Thr622 residue in the C1 domain was focused in this study for the interest in MD simulation. Actually, considering the motions of the C1 domain of the RhoA-bound WT-PARG1 (Fig. 5a) or I645V and T622M complexes (Fig. 9, see discussion), it is difficult to add to the study and discuss the mutational effects on the RhoA interactions in the conformational dynamics (Arg606 residue was not involv

---

## [Decision Letter · Decision Letter 1]

Dear Dr. Kouchi,

Thank you for submitting your manuscript to PLOS ONE. While the paper has improved by revision in some respects, there are still important questions to be addressed before the paper is ready for publication.

We look forward to receiving your revised manuscript.

Kind regards,

Paul A. Randazzo

Academic Editor

PLOS ONE

Reviewers' comments:

Reviewer's Responses to Questions

**Comments to the Author**

Reviewer #1: (No Response)

Reviewer #2: All comments have been addressed

2. Is the manuscript technically sound, and do the data support the conclusions?

Reviewer #1: No

Reviewer #2: Yes

3. Has the statistical analysis been performed appropriately and rigorously?

Reviewer #1: No

Reviewer #2: Yes

4. Have the authors made all data underlying the findings in their manuscript fully available?

Reviewer #1: Yes

Reviewer #2: Yes

5. Is the manuscript presented in an intelligible fashion and written in standard English?

Reviewer #1: Yes

Reviewer #2: Yes

Reviewer #1: The manuscript presents a computational analysis of the structural dynamics of PARG1, a RhoGAP protein, and its interactions with RhoA, focusing on the effects of the T622M and I845V mutations. The authors used a combination of structural modeling, molecular dynamics (MD) simulations, and binding free energy calculations (via MMPBSA) to investigate the recognition mechanisms of PARG1 and how these mutations disrupt its interaction with RhoA. The authors also explored the conformational changes induced by these mutations and provided insights into how these alterations may affect the GTPase activation process. Despite the novelty and relevance of the study, several methodological and interpretative concerns need to be addressed to strengthen the conclusions drawn and clarify aspects of the experimental approach.

Concerns:

o Clarify whether AlphaFold-Multimer was used for the PARG1-RhoA complex or just single-chain PARG1.

o Justify why AlphaFold was chosen and address potential limitations in modeling protein-protein interactions, especially if flexible loops like the C1 domain are involved.

o Discuss whether AlphaFold’s predictions were validated by experimental data (e.g., X-ray, cryo-EM) or alternative structural methods.

o Discontinuities in the RMSD graphs should be addressed—likely due to improper handling of periodic boundary conditions (PBC).

o The 2-µs simulation shows RMSD up to 10 Å, which is drastically higher than the 100-ns simulation (3 Å). Explain why both short and long simulations are compared and whether they are directly comparable.

o Ensure RMSD graphs are properly processed, including re-centering and imaging of the trajectories to handle PBC. Consider focusing on only one longer simulation (e.g., 2-µs MD) for better accuracy.

o Justify why specific frames (e.g., 15-30ns for WT PARG1, 15-45ns for p.Ile845Val) were selected for MMPBSA calculations.

o Clarify the criteria for selecting frames based on 2D-RMSD analysis and explain how these frames represent an equilibrated state of the system.

o The terms Pattern A and Pattern B are introduced but not sufficiently explained. Clarify what these patterns represent (e.g., specific conformational states, structural transitions, or interaction patterns).

o Provide a clear definition of Pattern A and Pattern B, along with how they relate to the simulation dynamics and protein-protein interactions.

o Consider focusing on the 2-µs MD simulation for better insight, as the shorter trajectory may not provide additional significant information.

o Integrate the biological significance of the mutations (T622M, I845V) more explicitly. Link the computational findings with experimental data or known functional defects in RhoGAP and RhoA regulation.

o Simplify and clarify figure legends, particularly explaining the meaning of labels like “1”, “2”, and “3” in the RMSD plots.

Reviewer #2: The reviewer thanks the author for addressing every concern previously raised and for the amount of work that that entailed. The manuscript is now ready for publishing, although with very minor alterations. The suggestions for those alterations are outlined here:

Response to General Comment 1: The term "Hydrophilic interface" is still used in a couple cases: figure legend for 3C and 3D; figure legend for 6B. Please make consistent by renaming "C-terminal interface."

Response to General Comment 4: A statement in the Discussion should be made indicating that, while not examined, it is plausible that the presence of the F-BAR domain and linker region to C1 domain may create steric constraints that cause preference for either pattern A or B as described in the text, or in other words, that the F-BAR domain may disrupt pattern A or B. There is no need to discuss the F-BAR more than that in this context, but it is important to note that its presence may alter the dynamics of the C1 domain relative to the GAP domain and that such a consideration was not in the scope of this work.

Figure 1E: Apologies that my last comment for this panel was not clearly written. Please make the C1 domain light blue as seen in panels 1F and 1H. RhoA is depicted in green in all other panels, so the C1 domain should not be green in panel 1E.

Figure 5B: One usage of "variable conserved" is shown, please change as this is not intuitive and was largely removed in the most recent version of the manuscript.

Figure 9: An admiral effort to summarize a lot of information! Could use slight improvement as it is still difficult to parse. For example, the color coding of C1 in WT (pattern B) is blue, but blue is also used to highlight the C1 domain in I845V at 960 ns. Consider making these separate panels (i.e., WT as panel A, I845V as panel B, and T622M as panel C) to reduce confusion. Finally, the summary in 9B (which should thus become panel D) should be created for I845V and T622M mutants (as panels E and F).

**Do you want your identity to be public for this peer review?** For information about this choice, including consent withdrawal, please see our Privacy Policy

Reviewer #1: No

Reviewer #2: No

---

## [Author Response · Author response to Decision Letter 2]

24 May 2025

Dr. Paul A. Randazzo

Academic Editor

PLoS ONE

Revision of manuscript, submission ID PONE-D-24-50037R1 [EMID:ea5600c191900474]

Dear Dr. Randazzo,

Please find attached revised manuscript and marked-up-copy of the manuscript that highlights changes made to the previously submitted manuscript by Kouchi and Kojima entitled “C1-linker region of PARG1 RhoGAP promotes the catalytic recognition fold of RhoA substrate“, as research article “PLoS ONE”. We read carefully your recommendations and the reviewer’s criticism and revised the paper accordingly. Here is our point-to-point response to the reviewer’s comments, experiment and discussion of all changes in the revised manuscript.

In revised manuscript, all the data obtained by 100-ns MD simulation were removed from Figure 4-9 and the structural analysis were performed from 2-�s MD simulation by adding the analysis for T622M-RhoA complex. The discussion of AlphaFold application to the GAP complex model and MD analysis for another model (Figure 1i in addition to Figure 1h) obtained by AlphaFold multimer were also added to the Discussion part in the revised manuscript. In the modeled structures using AlphaFold multimer indicated as Figures 1h and 1i, stable complexes were not obtained more than 50-75 ns through long-ranged simulation in the same conditions, presumably due to no interface between the C1 region and RhoA substrate formed. These irreversible dissociation patterns of the complex were not included in the result section, but deduced structural properties of the stable complexes obtained by MD simulation was included as the Figure S11 and described in the Discussion part. The summary of MD analysis for AlphaFold model from the aspects of RhoA interface will be helpful for understanding the dynamics of the GAP fold, and also support the notion that the GAP domain alone is not sufficient for the RhoA interaction. The criteria for selected structure complexes based on 2D-RMSD are included as Figure 4d-f and summary of the major conformational changes were included as Figure 9.

To Reviewer 1

The authors also explored the conformational changes induced by these mutations and provided insights into how these alterations may affect the GTPase activation process. Despite the novelty and relevance of the study, several methodological and interpretative concerns need to be addressed to strengthen the conclusions drawn and clarify aspects of the experimental approach.

A. Technical approach for the structural data and analysis was clarified in the Method and Result sections. Answers to the concerns for No. 4-6, 9-10, and 13 from reviewer 1 were not included in this form, because all the 100ns-scale MD simulation data were removed from the manuscript and figures, according to the concerns No.6 and 9 by reviewer 1.

Concerns

1. Clarify whether AlphaFold-Multimer was used for the PARG1-RhoA complex or just single-chain PARG1.

A. PARG1-RhoA complex was modeled using AlphaFold-Multimer and just single-chain PARG1 was modeled using AlphaFold and subjected to the HDOCK analysis.

The structural model selected for molecular dynamics analysis was based on template-free mode of HDOCK analysis, therefore structural data obtained from single-chain PARG1 was examined using the same mode of HDOCK program. In conclusion, the docking analysis of AlphaFold structure model of the single-chain was not consistent with the experimentally validated substrate preference of PARG1 reported by several papers including interactome analysis.

AlphaFold-multimer is one of the available complex analysis tools and selected top model 1-5 were checked, based on the similarity to crystallized RhoGAP-RhoA complexes (PDB: 5c2k, 3msa, 1ow3, 5irc) and interface analysis using RMSD and DrugscorePPI. Model1 and 3 obtained using AlphaFold-multimer (AF-MM) were selected by the similarity to the p190ARhoGAP (p50 and ArhGAP20) and MgcRacGAP, respectively. The description was added to Result section.

2. Justify why AlphaFold was chosen and address potential limitations in modeling protein-protein interactions, especially if flexible loops like the C1 domain are involved.

A. Several papers have reported that AlphaFold-mulimer performance was not good without multiple sequence alignment (Strom JM and Luck K (2025) Current Opinion in Struct. Biol. 91: 103002, Yin et al. (2022) Prot. Sci. 31: e4379). It has been pointed out that domain structures with intrinsic disordered domain (IDR) did not show good scores. We assessed the DockQ scores of the selected five AF-MM PARG1 structural models, and the calculated scores compared to the corresponding selected wild-type PARG1 model containing the C1-loop that was used for 2 micros MD analysis showed 0.345-0.359, with acceptable quality to the modeled structure selected using HDOCK with template-free mode. Detailed description for the reason was added to the Result section, pp. 12.

3. Discuss whether AlphaFold’s predictions were validated by experimental data (e.g., X-ray, cryo-EM) or alternative structural methods.

A. The predicted errors of regional GAP domain of AlphaFold model were mostly detected in the C-terminal alpha9 helix in the x-ray crystallized structures (in the case of 1ow3, 3msx, and 5irc as PDB ID). When compared to the complex models obtained from AlphFold-multimer to the crystallized complexes, the models with minimal RMSD value between 1ow3, 3msx, 5irc, and 5c2k showed higher percentage of cumulative binding free energies of conserved and variable interface residues than those with maximal RMSD value when calculated using DrugscorePPI program. In addition, the proximity of the interface residues of AlphaFold-multimer models were matched to the profiles of crystallized complexes as shown in Figure S5. Therefore, we analyzed model 1 and 3 of AlphaFold-multimer, selected based on RMSD values to these experimentally validated structures, for MD simulation. Detailed description was added to the Result section (pp. 12, 20) and Discussion part.

7. Justify why specific frames were selected for MMPBSA calculations.

A. In 2-micro s MD simulation, significant levels of conformational changes were detected and the trajectories were categorized as clusters based on the values of 2D-RMSD. The structural ensembles were classified into 2-8 clusters in each MD simulation based on the differences. Therefore, MMPBSA analysis was performed from the representative structure frames showing the minimal RMSD values within the largest cluster in each RhoA complexes, although the several representative structures were present in the structural ensembles. This reason was described in Material and methods and Result parts (p.15) with Figure 4.

8. Clarify the criteria for selecting frames based on 2D-RMSD analysis and explain how these frames represent an equilibrated state of the system.

A. selecting frames based on 2D-RMSD analysis was based on 8 clusters according to the RMSD value as shown in Figure 4. In Figure 9 the representative structures selected from several clusters showing significant conformational changes (i.e. RMSD value: 4-6) were depicted as the superimposed structures. The representative structures (state 2) were selected from the largest clusters during 2-�s MD simulation. Detailed description for producing 2D-RMSD matrices and selecting the representative structures was also added in Materials and Methods section.

9. Consider focusing on the 2-�s MD simulation for better insight, as the shorter trajectory may not provide additional significant information.

A. Because 100ns-trajectory data were not sufficient to provide structural variations in this case, we focused on 2-micro s MD simulation, and presented the results of MD data in Figure 4-9 and Table 1 in main text by removing the description on the short trajectory data in this revised manuscript.

12. Integrate the biological significance of the mutations (T622M, I845V) more explicitly. Link the computational findings with experimental data or known functional defects in RhoGAP and RhoA regulation.

A. No biological data has been reported for mutations presented in this manuscript. Therefore, we conducted molecular cloning of the corresponding C1-GAP region (residue number 611-886) of human PARG1 GAP domain from HEK293 cells. Unfortunately, the C1-GAP domain of PARG1 tagged with 3 tandem FLAG epitopes was not exogenously expressed in Neuro2a cells, although the tagged full-length MAGL (monoacylglyecerol lipase, positive control) coding a 33kDa protein was successfully expressed in the same condition (see the following paper as reference: Kouchi Z (2015) Biophys. Biochem. Res. Communication, 464, 603-610). As mentioned before in the previous letter, it was also reported that YFP-tagged PARG1 was poorly expressed in HEK293 cells, which was not sufficient level for calculation of luciferase assay in the following paper (Anderson EL and Hamann MJ, Biochem. J. 441: 869-871, 2012). The comment on the difficulties in cellular studies of PARG1 was included in the Discussion part of the revised manuscript. Related references on the variants/mutations were included as previously submitted.

To Reviewer 2

Response to General comment 1: The term “Hydrophilic interface” is still used in a couple cases: figure legend for 3C and 3D; figure legend for 6B. Please make consistent by renaming “C-terminal interface.”

A. The term “Hydrophilic interface” was corrected to the “C-terminal interface” in the figure legends

for Figures 3C and D, and 6B.

Response to General comment 4: A statement in the Discussion should be made indicating that, while not examined, it is plausible that the presence of the F-BAR domain and linker region to C1 domain may create steric constraints that cause preference for either pattern A or B as described in the text, or in other words, that the F-BAR domain may disrupt pattern A or B. There is no need to discuss the F-BAR more than that in this context, but it is important to note that its presence may alter the dynamics of the C1 domain relative to the GAP domain and that such a consideration was not in the scope of this work.

A. A statement for the effects of the presence of F-BAR domain was added to the end of the 6th paragraph in the Discussion part.

Fig.1E: Please make the C1 domain light blue as seen in panels 1F and 1H. RhoA is depicted in green in all other panels, so the C1 domain should not be green in panel 1E.

A. The C1 domain of Figure 1E is labeled in light blue as seen in panels 1F and 1H.

Fig. 5B: One usage of “variable conserved” is shown, please change as this is not intuitive and was largely removed in the most recent version of the manuscript.

A. Usage of “variable conserved” was removed from the revised manuscript in Figure 5b.

Fig. 9: Color coding of C1 in WT, I845V, and T622M complex to reduce confusion and the summary in 9B (which should thus become panel D) should be created for I1845V and T622 mutants (as panels E and F).

A. Color coding of C1 in WT, I845V, and T622M were unified with light blue for state 1 and 3, and blue for state2 that are the representative structure selected from the largest cluster in 2D-RMSD. The superimposed structures are separately indicated as Figure 9a-c. The summary panel focusing on the map of interface residues in the C1 domain and its residue-residue contact in the Glu723 residue located in �3-helix in each complex was created as Figure 9d-f in the revised manuscript.

---

## [Decision Letter · Decision Letter 2]

C1-linker region of PARG1 RhoGAP promotes the catalytic recognition fold of RhoA substrate

PONE-D-24-50037R2

Dear Dr. Kouchi,

We’re pleased to inform you that your manuscript has been judged scientifically suitable for publication and will be formally accepted for publication once it meets all outstanding technical requirements.

Kind regards,

Paul A. Randazzo

Academic Editor

PLOS ONE

Additional Editor Comments (optional):

Reviewers' comments:

Reviewer's Responses to Questions

**Comments to the Author**

Reviewer #1: All comments have been addressed

Reviewer #2: All comments have been addressed

2. Is the manuscript technically sound, and do the data support the conclusions?

Reviewer #1: Yes

Reviewer #2: Yes

3. Has the statistical analysis been performed appropriately and rigorously?

Reviewer #1: Yes

Reviewer #2: I Don't Know

4. Have the authors made all data underlying the findings in their manuscript fully available?

Reviewer #1: Yes

Reviewer #2: Yes

5. Is the manuscript presented in an intelligible fashion and written in standard English?

Reviewer #1: Yes

Reviewer #2: Yes

Reviewer #1: The authors have addressed all my comments and significantly revised the manuscript. I don't have any concerns.

Reviewer #2: The authors have addressed all comments as requested. Prior to full acceptance, the reviewer requests that the authors please proofread their manuscript thoroughly to ensure that all figures are cited properly (e.g., page 24, figure 9B is cited but it seems that perhaps 9D was supposed to be cited here?). Similarly, the figure legend for figure 9B exhibits a typo, in that the I845V mutant is described using "state 2" twice rather than listing "state 3" as well.

One concern is that some of the statistics seem to be arbitrary, e.g., the authors cite in figure legend of 6C that hydrogen bonds that are present more than 0.2% of the time are shown, but is this something significant worth showing? The same issue occurs in figure 5, where a 5% threshold is used. It seems possible that these thresholds were only chosen because of the prior analyses on the 100 ns (rather than 2 us) simulation time, so perhaps better justification is needed in some cases.

As a final comment, the authors indicate that their simulations of the C1-GAP domains of PARG1 are shown interacting with the inactive conformation of RhoA, rather than the active conformation. The justification for examining the inactive form of RhoA is not completely clear; could the authors comment more on why this state of RhoA was chosen instead of the active form, which is the substrate of the GAP?

**Do you want your identity to be public for this peer review?** For information about this choice, including consent withdrawal, please see our Privacy Policy

Reviewer #1: **Yes: ** Venkata Chirasani

Reviewer #2: No

---

## [Editor Report · Acceptance letter]

PONE-D-24-50037R2

PLOS ONE

Dear Dr. Kouchi,

I'm pleased to inform you that your manuscript has been deemed suitable for publication in PLOS ONE. Congratulations! Your manuscript is now being handed over to our production team.

Kind regards,

on behalf of

Dr. Paul A. Randazzo

Academic Editor

PLOS ONE